# Spectroscopic studies of atomic defects and bandgap renormalization in semiconducting monolayer transition metal dichalcogenides

Tae Young Jeong[1,2,5], Hakseong Kim[1,5], Sang-Jun Choi[3], Kenji Watanabe [4], Takashi Taniguchi[4], Ki Ju Yee[2], Yong-Sung Kim [1] & Suyong Jung [1]

Assessing atomic defect states and their ramifications on the electronic properties of two-dimensional van der Waals semiconducting transition metal dichalcogenides (SC-TMDs) is the primary task to expedite multi-disciplinary efforts in the promotion of next-generation electrical and optical device applications utilizing these low-dimensional materials. Here, with electron tunneling and optical spectroscopy measurements with density functional theory, we spectroscopically locate the mid-gap states from chalcogen-atom vacancies in four representative monolayer SC-TMDs—$WS_2$, $MoS_2$, $WSe_2$, and $MoSe_2$—, and carefully analyze the similarities and dissimilarities of the atomic defects in four distinctive materials regarding the physical origins of the missing chalcogen atoms and the implications to SC-mTMD properties. In addition, we address both quasiparticle and optical energy gaps of the SC-mTMD films and find out many-body interactions significantly enlarge the quasiparticle energy gaps and excitonic binding energies, when the semiconducting monolayers are encapsulated by non-interacting hexagonal boron nitride layers.

[1] Quantum Technology Institute, Korea Research Institute of Standards and Science, Daejeon 34113, Korea. [2] Department of Physics, Chungnam National University, Daejeon 34134, Korea. [3] Center for Theoretical Physics of Complex Systems, Institute for Basic Science, Daejeon 34126, Korea. [4] Advanced Materials Laboratory, National Institute for Materials Science, 1-1 Namiki, Tsukuba 305-0044, Japan. [5] These authors contributed equally: Tae Young Jeong, Hakseong Kim. Correspondence and requests for materials should be addressed to Y.-S.K. (email: yongsung.kim@kriss.re.kr) or to S.J. (email: syjung@kriss.re.kr)

Atomic defect states in two-dimensional semiconducting transition metal dichalcogenides (SC-TMDs) are credited to many intriguing scientific and engineering aspects, such as single quantum emitters[1–4], single-atom magnetism[5], and defect-modulating dopings[6,7], to name a few. Defect states have the implications to decide the specific type of SC-TMD, while defect-related trap states adjust the position of the Fermi level ($E_F$) with respect to conduction and valence bands[6–9]. The presence of atomic defects along with charged impurities is also known to limit charged carrier mobilities and induce a strong Fermi-level pinning at the metal/SC-TMD interfaces, thereby limiting SC-TMD-based electronic applications[10]. Thus, examining the origins and spectral locations of defect-induced states is prerequisite to the full utilization of such states in SC-TMDs and the building-up of TMD-based device functionalities. Experimental analyses of such defects typically rely on imaging methods, like transmission electron microscopy (TEM) and scanning tunneling microscope (STM), visualizing the various forms of atomic defects in SC-TMDs[6,7,10–13]. Although some STM studies have provided experimental signatures of the mid-gap states from atomic defects in SC-TMDs[14], the majority of outstanding questions, such as those concerning the defect-induced mid-gap states and their spectral locations inside the energy gaps, have mostly relied on theoretical predictions[8].

We note that spectroscopic investigations of defect-induced mid-gap states should be accompanied by the identification of other key SC-mTMD material parameters; the quasiparticle energy gaps ($E_g$) and the position of $E_F$ inside the gaps. For example, chalcogen-atom vacancies ($V_S$, $V_{Se}$), the most common defects in S-based and Se-based SC-TMDs, are expected to induce mid-gap states: singlet $a_1$ states forming close to the valence band and doublet $e$ states forming deep inside the energy gap[6,8]. Quite surprisingly, however, experiment and theory have yet to agree on the most fundamental SC-mTMD property, the quasiparticle energy gaps. For example, such energy gaps inferred from different experiments have varied up to $\approx 1\,\mathrm{eV}$[15], and theoretically expected energy gap sizes also differ depending on the extent of Coulombic interactions[15,16]. Accordingly, excitonic-binding energies ($E_B$), or the energy difference between quasiparticle and optical energy gaps ($E_{opt}$), are reported to vary from a few hundredth of meV to $\approx 1\,\mathrm{eV}$[17–21].

In this article, we carry out careful electron tunneling, optical spectroscopy measurements, and density functional theory (DFT) calculations for four representative SC-mTMDs—mWS$_2$, mMoS$_2$, mWSe$_2$, and mMoSe$_2$—to accurately assess material parameters, such as the mid-gap states from chalcogen-atom vacancies, the quasiparticle energy gaps and exciton-binding energies, and other material specifications. We are able to identify the mid-gap states from chalcogen-atom vacancies in the SC-mTMDs by introducing $h$-BN as a tunnel barrier and graphene as a spectrum analyzer in van der Waals (vdW)-based planar heterostructures. Defect-induced mid-gap states in the four respective films reveal similarities and dissimilarities regarding the spectral locations of the defect states, mechanisms of vacancy formation, and overall defect density of states (DOS). Our studies suggest that single-atom defects in SC-mTMDs present direct experimental implications that chalcogen-atom vacancies turn into active charged dopants. Moreover, we can accurately determine the quasiparticle energy gaps and exciton-binding energies of the four SC-mTMDs via electron tunneling, optical reflectance/transmission spectroscopy, and temperature-dependent photoluminescence (PL) measurements. We confirm that the electronic structures of the SC-mTMDs are greatly renormalized by electron–electron interactions. With temperature-dependent PL, we then assess the excitonic-binding energies of these monolayers to be as large as $\geq 0.78\,\mathrm{eV}$, when the films are encapsulated by $h$-BN and graphene.

## Results

**2D vdW planar tunnel junctions with SC-mTMDs.** An optical viewgraph and schematic of the vdW heterostructure for probing the electronic and optical properties of SC-mTMDs are displayed in Fig. 1a. Here, we list a few experimental highlights for implementing graphene as the bottom contact to a given SC-mTMD. First, direct metal contact to a SC-mTMD induces electronic and physical deformations on the film, which renders contact resistances comparable with or even larger than tunnel resistances through the $h$-BN barriers at low temperatures. This additional resistance along the path of tunnel electrons violates the ultimate priori for electronic tunneling spectroscopies, requiring that a major voltage drop occur at the tunnel junction in order for sample-bias voltage ($V_b$) to be associated with tunnel-electron energy with respect to the Fermi level ($E_F$). Thus, a reliable and low-resistance metal–SC-mTMD contact, which we have achieved via the bottom graphene, is critical for accurately addressing the electronic structures of SC-mTMDs.

The second advantageous role of the graphene bottom contact is that it allows the single-atom carbon layer to be used as a spectrum analyzer. As schematically illustrated in Fig. 1b, tunnel electrons injected from the graphite probe detect the electronic band structure of the SC-mTMD only if the tunnel electrons possess higher (lower) energies than the conduction (valence) band of the SC-mTMD. Otherwise, tunnel electrons exclusively detect the bottom graphene through the energy gap windows of both $h$-BN and SC-mTMD. Thus, any deviations in the spectra from the graphite–insulators–graphene tunnel junction, noted hereafter as the graphene baseline, should be attributed to the electronic structure of the SC-mTMD and the probable mid-gap states. The third role of the graphene is to minimize probe-induced charging effects. During tunneling measurements, the effective electric field between the probe and SC-mTMDs through the $h$-BN barrier becomes strong when $V_b$ increases up to $|V_b| \geq 1\,\mathrm{V}$. Thus, probe-induced charges and consequent electronic structure modifications in the SC-mTMD could cause serious complications in accurately analyzing tunnel spectra[22,23]. Thanks to a much larger charge compressibility of the graphene, however, induced charges accumulate only on the bottom graphene when $V_b$ is within the energy gaps, thereby allowing us to probe the intrinsic band structure of the SC-mTMD.

**Electron tunneling and optical spectroscopy measurements.** Figure 1d shows a collection of tunneling spectra, differential conductance ($G = dI/dV_b$) curves as a function of $V_b$ from mWS$_2$, mMoS$_2$, mWSe$_2$, and mMoSe$_2$ planar junctions at $T \leq 4\,\mathrm{K}$. In our scheme, $V_b$ is applied to a graphite probe with the SC-mTMD–graphene connected to a current preamplifier. Thus, negative (positive) $V_b$ corresponds to the empty (filled) states of the SC-mTMD at a positive (negative) energy with respect to $E_F$. The colored arrows in Fig. 1d, respectively, represent the locations of the quasiparticle energy gaps of the SC-mTMDs; detailed methods for locating the energy gaps will be described in later sections. The inset in Fig. 1d displays a graphene baseline $dI/dV_b$ with the Dirac point of graphene tuned at $V_b = 0\,\mathrm{V}$, and green lines indicate numerical fittings to experimental data (red circles)[22,24]. We compare several graphite–$h$-BN–graphene junctions and find out that all tunnel spectra maintain similar $dI/dV_b$ characteristics only differentiated by multiplication constants, predetermined by the tunnel $h$-BN thicknesses and junction areas (Supplementary Fig. 1).

Our planar heterojunctions allow us to address both electronic and optical SC-mTMD properties without switching device platforms. Figure 1e shows a collection of PL measurements at $T = 300\,\mathrm{K}$ and $T = 80\,\mathrm{K}$ from identical SC-mTMD-based planar

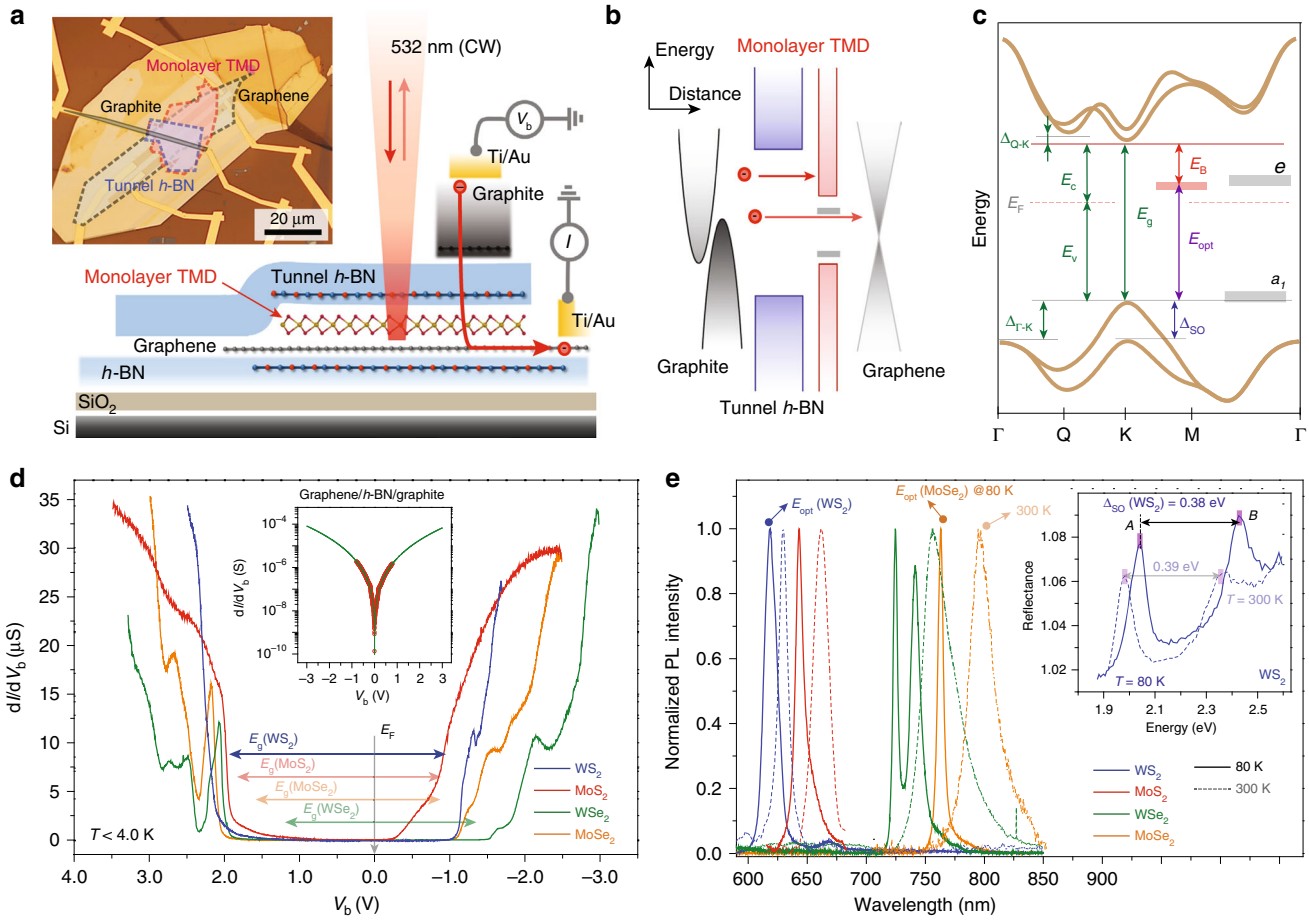

**Fig. 1** Electron tunneling and optical spectroscopy studies with SC-mTMD vdW heterostructures. **a** Schematic and optical viewgraph of our mTMD-based vdW planar heterostructure for electron tunneling and optical spectroscopy measurements. For electron tunneling studies, sample-bias voltage ($V_b$) is applied to the top graphite, and tunneling current through the $h$-BN and SC-mTMD is monitored at the bottom graphene. For optical spectroscopy measurements, a laser with either continuous 532 nm for PL or supercontinuum white sources for reflectance measurements is incident vertically on the mTMD planar tunnel device, and optical signals are collected with a ×50 objective lens. **b** Simplified energy-band alignments of our SC-mTMD-based planar tunnel junctions. Electrons injected from the graphite probe detect either the SC-mTMD or the bottom graphene layer depending on $V_b$ and its relative position with respect to the mTMD electronic structures. **c** Representative energy-momentum dispersion relations for SC-mTMD films depicted with key material parameters: atomic defect states ($a_1$, $e$), quasiparticle ($E_g$) and optical ($E_{opt}$) energy gaps, exciton-binding energy ($E_B$), and others. **d** Series of differential conductance ($G = dI/dV_b$) curves as a function of $V_b$ from mWS$_2$ (blue), mMoS$_2$ (red), mWSe$_2$ (green), and mMoSe$_2$ (orange) planar tunnel junctions at $T \leq 4$ K. Arrows, respectively, represent the locations of the quasiparticle energy gaps. Inset: a representative $dI/dV_b$ spectrum from one of the planar graphite–$h$-BN–graphene tunnel junctions. Green lines indicate numerical fittings to the experimental data (red circles). **e** Collection of PL spectra from the SC-mTMD planar devices at $T = 300$ K (dotted lines) and $T = 80$ K (solid lines). Inset: reflectance spectra measured at $T = 300$ K (dotted lines) and $T = 80$ K (solid lines) from the mWS$_2$ planar tunnel device

devices. The dominant PL peak, identified as an $A$ exciton that determines the optical energy gap of the particular SC-mTMD, is measured right at the tunnel junction where the monolayers are encapsulated by thin $h$-BN and graphite probe on top, and graphene and thick $h$-BN on the bottom (Supplementary Fig. 2). As previously reported[25,26] and confirmed from our devices, $A$-exciton peaks of SC-mTMDs blue-shift as temperature decreases in the range of 80 K $\leq T \leq$ 300 K. Following the Varshni relation[25], we extrapolate the $A$-exciton peak at $T = 0$ K and assign it as the $E_{opt}$ of the SC-mTMD encapsulated with graphene and $h$-BN, such as $E_{opt}$ (mWS$_2$) = 2.051 eV, $E_{opt}$ (mMoS$_2$) = 1.936 eV, $E_{opt}$ (mWSe$_2$) = 1.724 eV, and $E_{opt}$ (mMoSe$_2$) = 1.638 eV (Supplementary Figs. 3, 4).

We can independently measure the spin–orbit coupling (SOC)-induced valence-band splittings by locating the $A$ and $B$ excitonic peaks through optical reflectance and transmittance measurements (Supplementary Fig. 5). The inset in Fig. 1e shows reflectance spectra from the mWS$_2$ device at $T = 80$ K and $T =$

300 K. Both excitonic peaks are clearly identifiable, as well as $A–B$ exciton spacing; therefore, the SOC-induced valence-band splitting ($\Delta_{SO}$) is determined to be $\Delta_{SO}$ (mWS$_2$) $\approx 0.38$ eV. Note that $\Delta_{SO}$ is weakly dependent on temperature unlike the individual peak positions of $A$ and $B$ excitons, and $\Delta_{SO}$ is less susceptible to the dielectric environments. We prepare several SC-mTMDs on different substrates and find that the $A–B$ exciton spacings are consistent, within an uncertainty level of ±0.01 eV, with varying dielectric environments and temperatures (Supplementary Table 1). With optical measurements, therefore, we can determine the SOC-induced valence-band splittings of all four SC-mTMDs: $\Delta_{SO}$ (mWS$_2$) = 0.38 eV, $\Delta_{SO}$ (mMoS$_2$) = 0.15 eV, $\Delta_{SO}$ (mWSe$_2$) = 0.43 eV, and $\Delta_{SO}$ (mMoSe$_2$) = 0.20 eV (Supplementary Fig. 6).

**Assessing atomic defects of S-based SC-mTMDs.** Figure 2a and c displays tunneling spectra from mWS$_2$ and mMoS$_2$ devices, replotting $dI/dV_b$ (Fig. 1d) in log scale. As displayed in Fig. 2a,

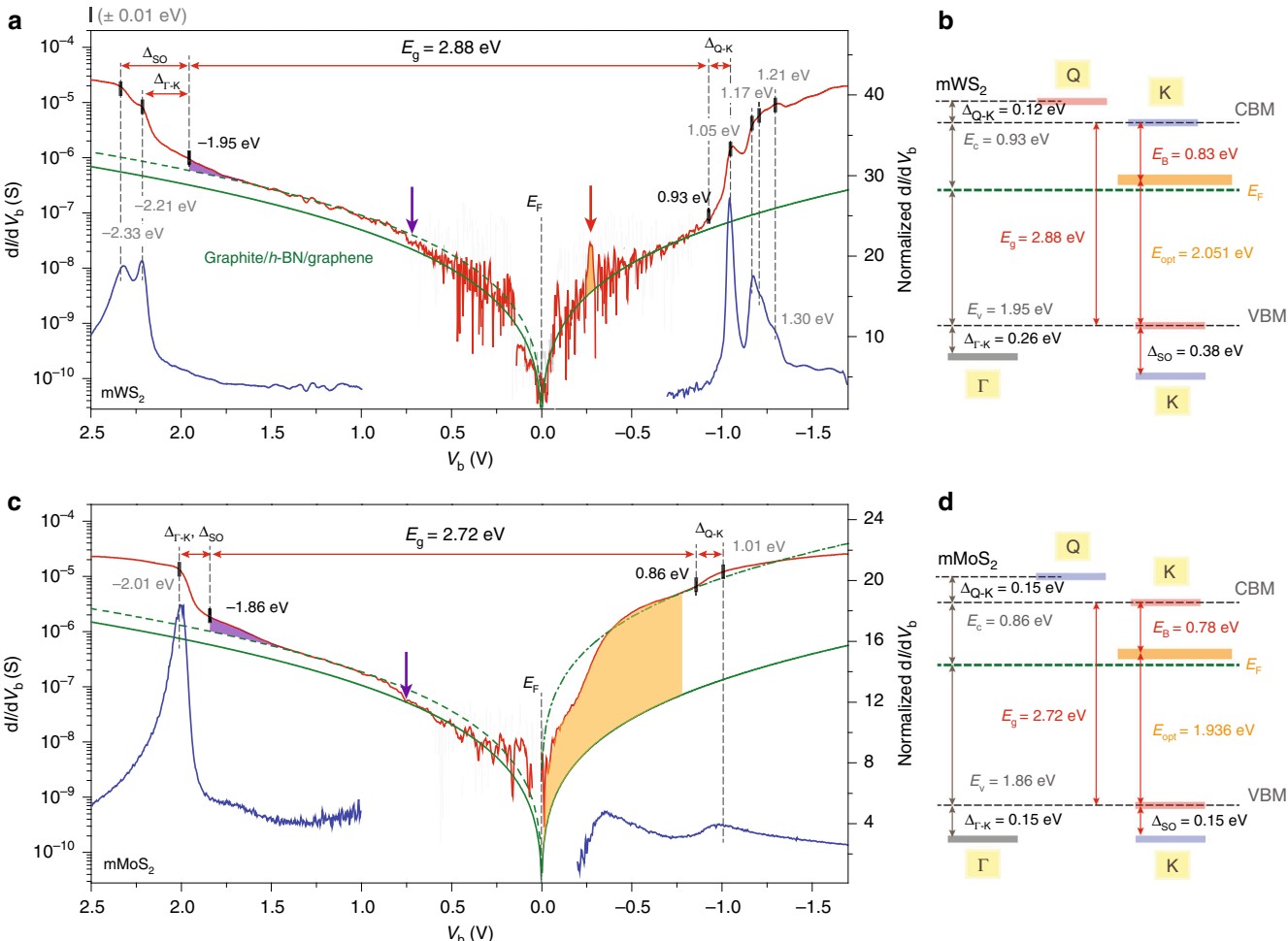

**Fig. 2** Detailed electronic structure analyses from electron tunneling spectroscopy measurements of S-based mTMDs. **a**, **c** Tunnel spectra plotted with $dI/dV_b$ in log scale from the $mWS_2$ (**a**) and $mMoS_2$ (**c**) planar tunnel junctions. Solid and dotted green lines mark the graphene baselines. Normalized conductance $(dI/dV_b)/(I/V_b)$ plots (blue solid lines) better represent tunneling spectra features at higher $V_b$. The areas delineated with purple and orange, respectively, represent the $a_1$ and $e$ defect states from sulfur-atom vacancies. $dI/dV_b$ spectral locations and key electronic structure assignments for the $mWS_2$ (**a**) and $mMoS_2$ (**c**) films are marked with dotted black lines and solid red arrows. **b**, **d** Summarized energy-level assignments for the $mWS_2$ (**b**) and $mMoS_2$ (**d**) films. Uncertainty for each energy-level assignment is less than ±0.01 eV

spectral features at higher $V_b$ are better identified with normalized conductance $(dI/dV_b)/(I/V_b)$ plots. To better distinguish $dI/dV_b$ at lower $V_b$, we numerically average out the original data and plot the leveled signals (red lines). Note that signals where $V_b$ is inside the $mWS_2$ energy gap ($|V_b| \leq 1$ V) reveal distinct $dI/dV_b$ variations with more than three orders of magnitude, and these spectra reflect the graphene baseline with both $h$-BN and SC-mTMD as tunnel insulators. Interestingly enough, the graphene baseline and the $mWS_2$ tunnel signals become well aligned in a $V_b$ range of $|V_b| < 0.7$ V for both filled ($V_b > 0$ V) and empty ($V_b < 0$ V) states. After a slight $dI/dV_b$ hike at $V_b \approx 0.7$ V, the $mWS_2$ spectra follow the graphite–graphene junction characteristics up to $V_b < 1.5$ V with a minor adjustment at the tunneling constant (dotted green line).

As discussed previously, any deviations from the graphene baseline can be directly related to the SC-mTMD film and its electronic structure. At first, we assign the $dI/dV_b$ peaks at −2.33 eV and −2.21 eV to the valence-band edges of the K and Γ points of the $mWS_2$, respectively. In $mWS_2$, the valence-band edge at Γ is expected to be higher in energy than the lower edge of the SOC-split valence band at K. Directly inferred from the optically determined $\Delta_{SO} (mWS_2) = 0.38$ eV, the higher SOC-split valence band, thus the valence-band edge of $mWS_2$, should be positioned

at −1.95 eV, where we locate a distinct $dI/dV_b$ tunnel feature (Supplementary Fig. 7). Following the same approach, the conduction-band edge at K is assigned at 0.93 eV, and the $dI/dV_b$ peak at 1.05 eV as the conduction-band edge at the Q point. Accounting for all these assignments, the $mWS_2$ film encapsulated with nonperturbing high-quality $h$-BN and graphene is confirmed to be a direct bandgap $n$-type semiconductor, whose conduction band is closer to $E_F$ than the valence band edge, and with a quasiparticle energy gap $E_g = 2.88$ eV, excitonic-binding energy $E_B = 0.83$ eV, and an optical energy gap $E_{opt} = 2.051$ eV. The uncertainty for each energy-level assignment is less than ± 0.01 eV. Complete energy-level alignments for $mWS_2$ are summarized in the diagram in Fig. 2b. We point out that the measured quasiparticle energy gap of $mWS_2$ is in perfect agreement with theoretical expectations from the GW approach considering many-body perturbation effects.

We now address the $mMoS_2$ electronic band structures following the above-discussed spectra-analyzing protocol with graphene baseline. As displayed in Fig. 2c, the graphene baseline follows most of the $dI/dV_b$ features from the $mMoS_2$ device in the filled states at $V_b < 1.5$ V. It is intriguing to note that, identical to $mWS_2$, a slight $dI/dV_b$ increase in value is also present at $V_b \approx 0.7$ V in $mMoS_2$. Our measurements indicate that the valence band

edge at $\Gamma$ is in alignment with the lower SOC-split band edge at the K point at –2.01 eV, and the $dI/dV_b$ feature at –1.86 eV where the tunnel spectra in log scale evidently changes its slope is assigned as the higher SOC-split valence-band edge at K (Supplementary Fig. 7). The valence-band splitting at K is in perfect agreement with optically confirmed $\Delta_{SO}$ (mMoS$_2$) = 0.15 eV.

Analyzing the mMoS$_2$ $dI/dV_b$ in the empty states, however, is not as straightforward as for mWS$_2$ or the filled states of mMoS$_2$. At first, $dI/dV_b$ in the empty states are significantly larger than the values in the filled states, by more than two orders of magnitude, making mMoS$_2$ $dI/dV_b$ highly asymmetric around $E_F$. Interestingly though, $dI/dV_b$ becomes realigned with the graphene baseline at $V_b \approx$ –0.8 V with a significantly enhanced $dI/dV_b$ multiplication constant. With that, we assign the conduction band edge of the mMoS$_2$ at the K point at 0.86 eV, and the $dI/dV_b$ peak at 1.01 eV as the band edge at Q. As a result, mMoS$_2$ can be considered as a direct bandgap $n$-type semiconductor, with a quasiparticle energy gap $E_g$ = 2.72 eV and excitonic-binding energy $E_B$ = 0.78 eV ($E_{opt}$ = 1.936 eV). Again, the experimentally assessed quasiparticle energy gap of the mMoS$_2$ is in excellent agreement with the predicted energy gaps from GW calculations[15,27].

We relate the enormously amplified $dI/dV_b$ in the empty states to the doublet $e$ defect states from sulfur-atom vacancies in the mMoS$_2$. Based on our DFT calculations, the majority of singlet $a_1$ states in mWS$_2$ are formed at lower energies than the valence-band edge, concealing most of the defect states under the valence band (see below). The bulk portion of $a_1$ states in mMoS$_2$, however, are expected to form at higher energies than the valence-band edge, expecting rather pronounced $a_1$-related mid-gap states. Defect states relating to the doublet $e$ states are formed deep inside the energy gaps with the mid-gap states in mMoS$_2$ closer to $E_F$ in the empty states. Our measurements confirm these theoretically expected spectral locations of $a_1$ and $e$ defect states in S-based SC-mTMDs. We specify that the $dI/dV_b$ regions, positioned higher in energy than the valence-band edges, therefore within the energy gaps with $dI/dV_b$ straying from the graphene baseline (depicted in purple in Fig. 2a, c), reflect the $a_1$ defect states. It is obvious that the $a_1$ states in mMoS$_2$ extend further into the gap when compared with the smaller $dI/dV_b$ region for the mid-gap states in mWS$_2$. In addition, an evident $dI/dV_b$ hump inside the energy gap of mWS$_2$, as marked with a red arrow and delineated in orange in Fig. 2a, is assigned to the $e$ defect states, and their spectral locations are consistent with theoretical calculations (see below). We further note that the defect-related $dI/dV_b$ features are not sensitive to external $V_g$, unlike to spectra relating to underlying graphene and its energy-band alignment with a graphite probe (Supplementary Fig. 8). As compared with the well-isolated $e$ states in mWS$_2$, however, the mid-gap states in mMoS$_2$ extend much wider inside the energy gap, making the defect-related tunnel spectra more conspicuous. The enlarged $dI/dV_b$ in the empty state suggests that mMoS$_2$ films are inclined to contain more sulfur vacancies than mWS$_2$, and that some of the vacancies are mobile enough to agglomerate into energetically stable line-type defects, as theoretically expected[13] and experimentally confirmed with TEM analyses[28]. In an earlier STM and our theoretical calculations confirm that defect states of neutral disulfur vacancies ($V_S V_S$), the simplest form of vacancy chains in mMoS$_2$, have $e$–$e$ divacancy hybridized defect states that are extensively expanded inside the energy gap[29].

**Assessing atomic defects of Se-based SC-mTMDs.** Figure 3a and c, respectively, displays the $dI/dV_b$ from mWSe$_2$ and mMoSe$_2$ planar devices. Note that tunnel $dI/dV_b$ structures in mWSe$_2$ are notably lacking around the valence band, and the peak at –2.05 eV and the location where tunnel spectra start deviating

from the graphene baseline are quite distant considering the SOC-split valence-band edges, $\Delta_{SO}$ (WSe$_2$) = 0.43 eV. We attribute these smooth $dI/dV_b$ evolutions to an augmented momentum mismatch of the tunneling electrons from the graphite probe to the mWSe$_2$ film in this particular tunnel junction. As discussed in previous reports, tunneling spectra from vdW planar heterojunctions are not only sensitive to the sample DOS at a given energy but also rely on the crystal momentum alignments of the tunnel probe (graphite) and the 2D layers of interest (SC-mTMD)[30–33]. When electrons with momentum $k_{gp}$ are injected from the graphite probe to the available states of the SC-mTMD at $k_{mTMD}$, momentum of the tunnel electrons relaxes by $\Delta k_{||} = |k_{mTMD} – k_{gp}|$, which is then directly linked to the probability of electron tunneling. The absolute value of tunnel $dI/dV_b$ is determined by the tunneling decay constant $T \approx (2m_e\Phi_b/\hbar^2 + (\Delta k_{||})^2)^{1/2}$, where $\Phi_b$ is the tunnel barrier and $m_e$ is the effective mass of tunnel electrons. Thus, a larger momentum mismatch ($\Delta k_{||}$) of tunnel electrons results in weaker signals and a smaller $dI/dV_b$ in value. Graphite is naturally considered as the most effective tunneling probe for investigating the electronic structures of graphene and SC-mTMDs since most of the interesting features of 2D hexagonal lattice materials are confined to the electronic structures around the K point, at which the Fermi level of the graphite probe is precisely located[34]. Here, we need to assert that only parts of the Brillouin zones (BZ) of the graphite probe and SC-mTMDs become matched, even for perfectly aligned planar junctions, because of the lattice mismatch between graphene and SC-TMDs (Supplementary Fig. 9). However, electron tunneling through 2D vdW heterojunctions consisting of layered materials with different lattice constants needs to consider extended BZs of the heterostructures and their varying equipotential surfaces depending on $V_b$. We find that both of which are sensitive to the misalignment angle of the junctions, and tunneling probability becomes highest for perfectly aligned heterostructures, which will be discussed in detail in coming literature.

We fabricate a second type of mWSe$_2$ tunnel device with crystalline angles of top graphite and mWSe$_2$ layers tightly aligned in order to efficiently probe the electronic structures around K. As expected, $dI/dV_b$ around the valence-band edge becomes enhanced in value in the tightly aligned device (solid orange line in Fig. 3a). Resonant tunnel features with negative $dI/dV_b$ (green circles in Fig. 3a) confirm that the crystalline angles of the graphite and mWSe$_2$ are closely matched[30–33]. By comparing the $dI/dV_b$ from aligned and misaligned devices, we assign the upper edge of the SOC-split valence bands at –1.21 eV, and the lower SOC-split band edge to a $dI/dV_b$ crest at –1.64 eV, guided by optically determined $\Delta_{SO}$ (mWSe$_2$) = 0.43 eV. The $dI/dV_b$ peak at –2.05 eV is subsequently designated as the valence-band maximum at $\Gamma$.

It is interesting to note that tunnel signals from the tightly aligned device in the vicinity of the conduction-band edge differ from the amplified $dI/dV_b$ around the valence band: $dI/dV_b$ at $\approx$1.35 eV are not amplified as much as those around the valence-band maximum at K. Instead, a couple $dI/dV_b$ peaks become conspicuous at 1.50 eV and 1.53 eV in the tightly aligned sample, leading us to assign them as the SOC-split conduction-band minima at the K point, and the band edge at 1.35 eV to the conduction-band minimum at Q (Supplementary Fig. 10). With these assignments, we assert that mWSe$_2$ is an indirect bandgap and weakly doped $p$-type semiconductor, whose quasiparticle energy gap is as large as $E_g$ = 2.56 eV and exciton-binding energy $E_B$ = 0.82 eV ($E_{opt}$ = 1.724 eV)[15,27]. Furthermore, SOC-split conduction bands at K are found to be $\Delta_{SO,C}$ (mWSe$_2$) = 0.03 eV. Recently, Wang et al. reported the similar value ($\Delta_{SO,C}$ (mWSe$_2$) $\approx$ 0.04 eV) by optical reflectance and PL spectroscopy measurements[35].

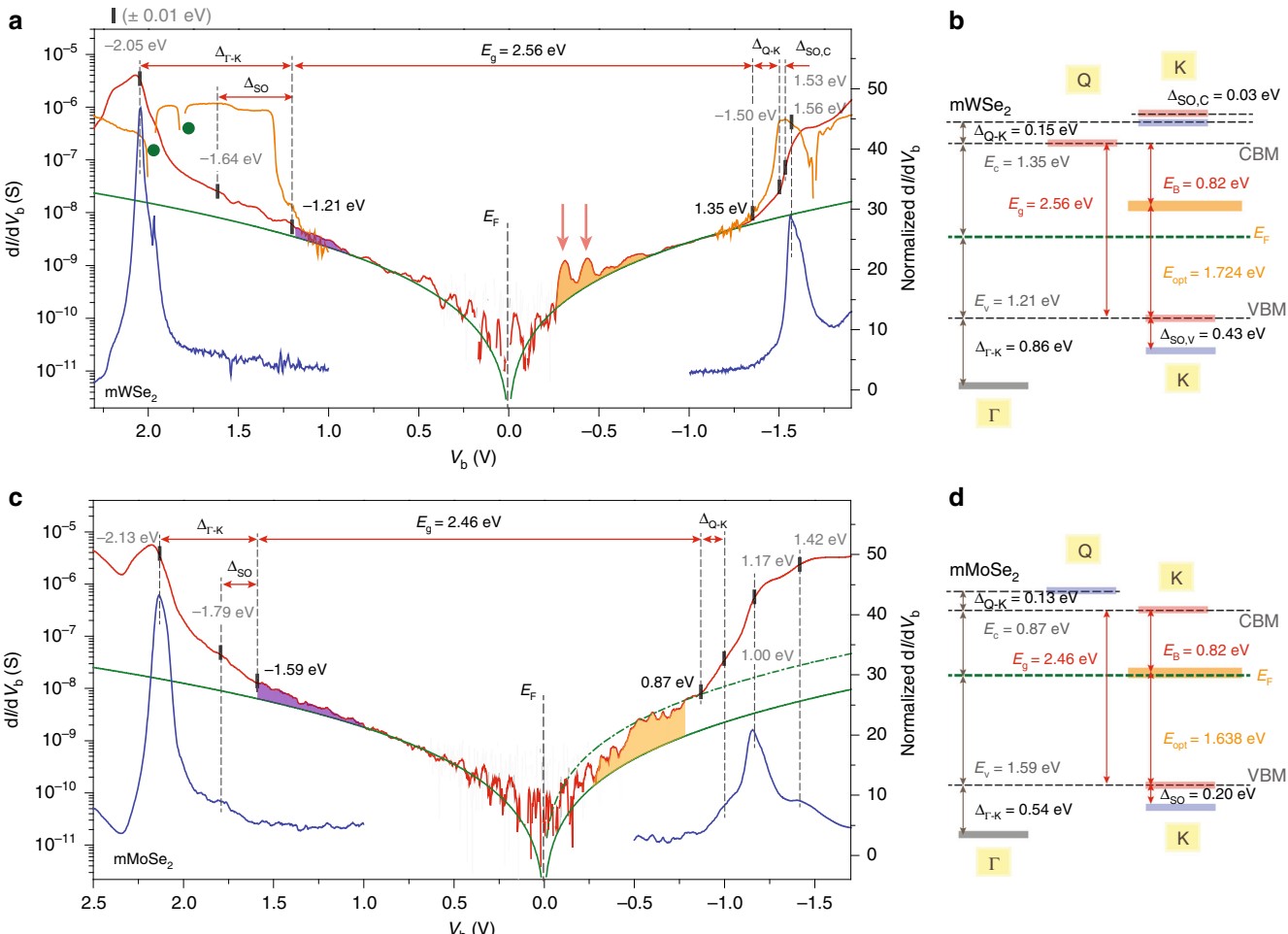

**Fig. 3** Detailed electronic structure analyses from electron tunneling spectroscopy measurements of Se-based mTMDs. **a**, **c** Tunnel spectra plotted with $dI/dV_b$ in log scale from the mWSe$_2$ (**a**) and mMoSe$_2$ (**c**) planar tunnel junctions. Tunnel spectra from the crystallographically aligned device are delineated with solid orange lines, and negative $dI/dV_b$ features for resonant tunneling are marked with green circles. Solid and dotted green lines indicate the graphene baseline spectra. The areas delineated with purple and orange, respectively, represent the $a_1$ and $e$ defect states from selenium-atom vacancies. **b**, **d** Summarized energy-level assignments for the mWSe$_2$ (**b**) and mMoSe$_2$ (**d**) films. Uncertainty for each energy-level assignment is less than ±0.01 eV

Although the rather moderate $dI/dV_b$ progression of the mMoSe$_2$ film is similar to the spectrum from the misaligned mWSe$_2$ device, a tunnel feature corresponding to the SOC-split lower valence-band edge at K is readily identified at −1.79 eV in both regular and normalized $dI/dV_b$ (Fig. 3c). Consequently, the higher SOC-split valence-band edge can be assigned at −1.59 eV based on the optically obtained $\Delta_{SO}$ (mMoSe$_2$) = 0.20 eV, and the $dI/dV_b$ crest at −2.13 eV marks the valence-band maximum at Γ. Notably, the mMoSe$_2$ spectra in the empty states are significantly enhanced in value similar to the mMoS$_2$, and $dI/dV_b$ becomes realigned to the graphene baseline at ≈0.8 eV with a sizable adjustment of tunneling constant (dotted green line in Fig. 3c). We assign the conduction-band minimum of the mMoSe$_2$ at 0.87 eV, which makes mMoSe$_2$ a direct bandgap $n$-type semiconductor with a quasiparticle energy gap $E_g$ = 2.46 eV and an exciton-binding energy $E_B$ = 0.82 eV ($E_{opt}$ = 1.638 eV).

The most common atomic defects in mWSe$_2$ and mMoSe$_2$ films are selenium-atom vacancies[12]. Similar to the sulfur vacancies in mWS$_2$ and mMoS$_2$, singlet $a_1$ states are expected to form closer to the valence bands with varying spectral locations in the Se-based TMDs; $a_1$ states in mWSe$_2$ mostly form below the valence-band maximum, while the bulk portion of mMoSe$_2$ $a_1$ defect states are expected to form inside the energy gap. Augmented $dI/dV_b$ attributed to the $a_1$ defect states are delineated

in purple in Fig. 3a and c, where the $a_1$ defect states in mMoSe$_2$ extend further inside the energy gap while the area relating to the mWSe$_2$ $a_1$ defect states is apparently narrower. Doublet $e$ defect states from the missing selenium atoms are expected to form mid-gap states inside the energy gaps, with corresponding tunnel spectra of mWSe$_2$ identified from the $dI/dV_b$ bumps inside the energy gap (red arrows in Fig. 3a, Supplementary Fig. 8). The amplified and extended $dI/dV_b$ of mMoSe$_2$, over which the $e$ defect states are known to exist, suggest that selenium vacancies are more prevalent in mMoSe$_2$ than mWSe$_2$, and some of them tend to form vacancy chains or clusters.

**Comparison of the atomic defect states in SC-mTMDs.** We numerically calculate the atomic defect DOS from chalcogen-atom vacancies in the four SC-mTMDs with a consideration of the SOC effect, and present overlaid plots with experimental data in Fig. 4a. In these calculations, we do not consider Coulombic many-body effects since a rather simplified DFT is sufficient enough to explain the atomic defect states in SC-mTMDs. It is crucial to note that the SOC effect influences not only the intrinsic electronic structures but also the defect states from chalcogen-atom vacancies[36]. As displayed in Fig. 4a, the SOC effect splits $e$ defect states by ≈0.2 eV in mWS$_2$ and mWSe$_2$, while

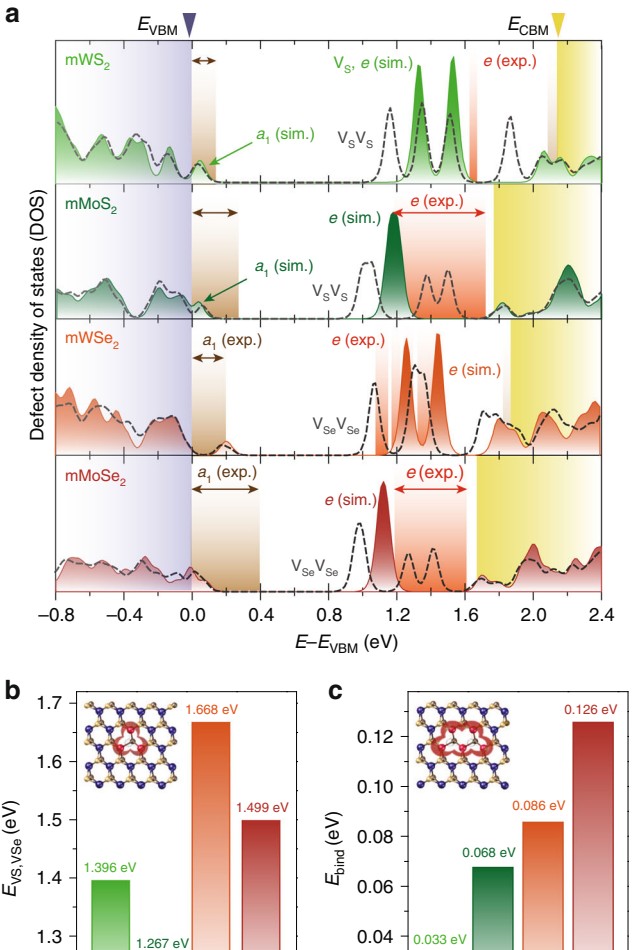

**Fig. 4** DFT studies on chalcogen-atom vacancies in SC-mTMDs. **a** DFT-calculated locations and density of states of charge neutral $a_1$ and $e$ defect-states from single chalcogen-atom vacancies in the four SC-mTMD films with a consideration of the SOC effect. The defect DOS induced by double chalcogen-atom vacancies are overlaid with dotted gray lines. Tops of the valence-band edges are set to 0 eV. Note that quasiparticle energy gaps from DFT calculations are severely underestimated without the consideration of electron–electron interactions. Experimentally identified $a_1$ and $e$ defect states are, respectively, overlaid with brown and orange shadows. **b**, **c** Formation ($E_{VS,VSe}$, **b**) and binding ($E_{bind}$, **c**) energies of individual chalcogen-atom vacancies in the four representative films

in our calculations, and find that the calculated spectral locations of both $a_1$ and $e$ states in mMoS$_2$, mMoSe$_2$, and mWS$_2$ films are consistently shifted to lower energies than the experimentally identified positions, with the sole exception being the $e$ defects in the weakly $p$-doped mWSe$_2$. Such consistent energy-level shifting of $a_1$ and $e$ defect states to higher energies (toward conduction bands) presents a strong experimental implication that chalcogen-atom vacancies in SC-mTMDs can work as dopants, especially as dominant donor states in mWS$_2$, mMoS$_2$, and mMoSe$_2$ films.

Here, it is worth mentioning that there could exist several other defects in the SC-mTMDs, such as H$_2$ or N$_2$ adatoms, transition metal-atom vacancies, or others. However, experimental identifications of such scant defects are daunting, especially in our planar junctions where active areas are as large as several microns in dimension. The observed mid-gap states are ensembles of all the available defects, whose spectra could be easily overshadowed by the more abundant defect sources; missing chalcogen atoms in S-based and Se-based SC-mTMDs. Moreover, the formation energies of chalcogen-atom vacancies are favorable to many other defects. For example, much higher formation energies of transition-metal-atom vacancies suggest that the missing transition-metal atoms are not energetically favorable to chalcogen-atom vacancies, not to mention that the spectral locations of such defects are different in energy from those of chalcogen-atom vacancies (Supplementary Fig. 12)[8,11,37,38]. In addition, we find that the binding energies of H on SC-TMDs are negative with respect to free H$_2$ molecules; –1.9, –2.1, –2.2, and –2.4 eV/H, respectively, for MoS$_2$, MoSe$_2$, WS$_2$, and WSe$_2$, as similar to N$_2$ chemisorption[37,38], suggesting that H atoms are more likely to desorb from the TMDs and form H$_2$ molecules instead. The calculated binding energies of H$_2$ on the surface of TMD films are a few tens of meV, which allow H$_2$ to be easily detached from the SC-TMD films as well.

To further clarify the similarities and dissimilarities regarding atomic defect states in SC-mTMDs, we calculate the formation energies ($E_{VS,VSe}$) of individual chalcogen-atom vacancies in the four films (Fig. 4b). Our calculations suggest that sulfur atoms are relatively easier to be taken off with lower atomic-defect formation energies than selenium atoms, implying that mMoS$_2$ ($E_{VS}$ (mMoS$_2$) = 1.267 eV) would have the most chalcogen-atom vacancies in total, while mWSe$_2$ ($E_{VSe}$ (mWSe$_2$) = 1.668 eV) would be the least vulnerable to losing selenium atoms. Moreover, we can deduce the density of states of chalcogen-atom vacancies in the intrinsic SC-mTMDs, revealing that mMoS$_2$ has the highest defect DOS with $D_{VS}$ (mMoS$_2$) = 1.46 × 10$^{12}$ cm$^{-2}$, followed by mWS$_2$ ($D_{VS}$ (mWS$_2$) = 6.95 × 10$^{11}$ cm$^{-2}$), mMoSe$_2$ ($D_{VS}$ (mMoSe$_2$) = 3.51 × 10$^{11}$ cm$^{-2}$), and finally mWSe$_2$ with the lowest DOS at $D_{VS}$ (mWSe$_2$) = 1.31 × 10$^{11}$ cm$^{-2}$.

Intriguingly, tunneling spectra related to the doublet $e$ defect states in mWS$_2$ remain as an isolated d$I$/d$V_b$ hump (Fig. 2a), despite the fact that mWS$_2$ layers are expected to have a larger defect DOS than mMoSe$_2$, in which a much extended defect-induced d$I$/d$V_b$ bump is formed (Fig. 3c). Our DFT calculations confirm that sulfur vacancies in mWS$_2$ films are the least likely to hybridize to form divacancies or vacancy chains, since the vacancy binding energy of mWS$_2$ is the lowest ($E_{bind}$ (mWS$_2$) = 0.033 eV) among the four films (Fig. 4c). In comparison, selenium-atom vacancies in mMoSe$_2$, with a vacancy binding energy ($E_{bind}$ (mMoSe$_2$) = 0.126 eV) one order higher than mWS$_2$, are expected to form line-type defects or defect clusters with relative ease, leading the mid-gap spectra originating from doublet $e$ defect states to extend wider within the energy gap. Finally, a few d$I$/d$V_b$ humps for the doublet defect states in the mWSe$_2$ (Fig. 3a) could imply that some selenium-atom vacancies become hybridized, justified by a relatively high vacancy binding

the splitting is minimal in mMoS$_2$ and mMoSe$_2$ (Supplementary Fig. 11). As experimentally and theoretically confirmed, the $a_1$ defect states of the Mo-based films extend further into the energy gap, while the majority of the $a_1$ states of the W-based films hide under the valence bands. Spectral locations of the doublet $e$ defect states in mWS$_2$ and mWSe$_2$ are close to the theoretical expectations, and calculated $e$ states in mMoS$_2$ and mMoSe$_2$ are in the vicinity of the expanded defect-induced d$I$/d$V_b$. We additionally calculate the defect DOS by double chalcogen-atom vacancies ($V_SV_S$ or $V_{Se}V_{Se}$) and present them with dotted gray lines in Fig. 4a. As noted previously, the $e$–$e$ divacancy defects become hybridized and form additional mid-gap states inside the energy gap, further extended toward the conduction-band edge in SC-mTMDs. We consider charge neutral chalcogen-atom vacancies

energy ($E_{bind}$ (mWSe$_2$) = 0.086 eV), although the overall number of selenium-atom vacancies is expected to be the lowest among the four SC-mTMDs. However, we should note that these d$I$/d$V_b$ bumps could be also attributed to the SOC-induced defect-state splittings, not just the double Se vacancies. At this moment, we cannot say with certainty whether the SOC or divacancy defects or both are the sources for the isolated set of d$I$/d$V_b$ bumps in mWSe$_2$, which calls for further theoretical and experimental works.

## Discussion

By implementing electron tunneling and optical spectroscopy measurements with DFT analyses, we have provided the most accurate and reliable material parameters to date of four representative SC-mTMD films. Specifically, our work includes spectra profiles of the mid-gap states from individual chalcogen-atom vacancies, quasiparticle energy gaps, and exciton-binding energies; all observed key material parameters are summarized in the Table 1. Through such experimental and theoretical studies, we are able to confirm that chalcogen-atom vacancies are the most prevalent atomic defects in SC-mTMDs, and these defects induce two mid-gap states around valence band edges (singlet $a_1$ states) and inside energy gaps (doublet $e$ states). Atomic defect states reveal similarities and dissimilarities among these four distinctive SC-mTMDs regarding the spectral locations of the defect-induced mid-gap states, physical vacancy formations, and intrinsic defect DOS. Our studies reveal that S-based mTMDs are more vulnerable to chalcogen-atom vacancies than their Se-based counterparts, presenting strong experimental implications that such vacancies are directly related to charged dopings in SC-mTMDs: S-based films are heavily doped $n$-type semiconductors, while Se-based films configure either as a moderately doped $n$-type semiconductor (mMoSe$_2$) or a weakly doped $p$-type (or intrinsic) semiconductor (mWSe$_2$). Competing with the overall defect DOS and vacancy binding energies, chalcogen-atom vacancies in the intrinsic mMoS$_2$, mMoSe$_2$, and mWSe$_2$ films are inclined to form hybridized vacancy chains while sulfur-atom vacancies in mWS$_2$ films remain isolated. In addition, we confirm that the energy bands of the films are greatly renormalized by enhanced many-body interactions, and diminished screening effects in the atomically thin 2D structures, leading to exceptionally large excitonic-binding energies up to ≥0.78 eV. By implementing crystalline angle-dependent electron-tunneling spectroscopies, we have demonstrated that tunneling decay constants, and therefore tunnel signals in the 2D vdW heterostructures, can be readily tuned by controlling the tunnel-electron momentums between tunnel probe and SC-mTMD crystals. This provides a useful experimental knob for investigating the electronic structures of various TMDs with much improved measurement accuracy. We believe that the key material parameters presented in this report can provide a solid foundation for current and next-generation electronic and optical applications with ultrathin semiconducting TMD films. Moreover, our experimental approaches can be applicable to any low-dimensional quantum materials and their unlimited combinations for high-precision material metrology.

## Methods

**Device fabrication.** In our planar vdW heterostructures, preparation of atomically clean interfaces is of critical importance for accurate and reliable material characterization of SC-mTMD films. At first, 60 -nm to 100- nm-thick high-crystalline $h$-BN flakes are mechanically exfoliated on a 90 -nm-thick SiO$_2$ layer on Si substrate. Prior to exfoliation, we thoroughly clean the SiO$_2$/Si substrates with acetone and IPA in an ultrasonication bath and subsequent dipping in piranha solution. We carefully examine $h$-BN surface cleanness with a dark-filtered optical microscope to avoid cracks and nonuniform $h$-BN layers. Then, single-layer graphene mechanically isolated onto polymer stacks of PMMA (poly(methyl methacrylate))–PSS (polystyrene sulfonate) layers is transferred to the prelocated $h$-BN flake on SiO$_2$/Si substrate using a dry transfer method. The total thicknesses of PMMA and PSS films on bare Si substrates are adjusted for optimal optical contrasts to identify the layer number of graphene and tunnel $h$-BN. Once the PMMA/PSS/Si substrates with 2D layers of graphene, $h$-BN, SC-mTMDs, and graphite on the polymer stacks are placed on top of DI water, the water soluble PSS layer is quickly dissolved, and the PMMA layer with 2D materials becomes isolated from the Si substrate. Then, with a micromanipulation stage equipped with a rotator and an optical microscope, we transfer the 2D layers on PMMA to a targeted location on a SiO$_2$/Si substrate with micrometer accuracy. After dissolving PMMA in warm acetone (60 °C), we further anneal the samples in forming gases of Ar and H$_2$ (9:1 ratio by flow rate). We set the annealing temperature at 350 °C for single-layer graphene and then reduce the temperature to 250 °C for SC-mTMD films to avoid undesired defect-state formations. After annealing, we confirm the surface cleanness with an atomic force microscope. By following a similar dry transfer protocol, SC-mTMD films, thin $h$-BN (3–4 layers), and graphite flakes are sequentially transferred on top of the graphite-$h$-BN stack to complete the mTMD-based planar tunneling device. For the second type of planar tunnel devices, we intentionally align the crystalline angles of the top graphite and monolayer WSe$_2$. No careful alignments are necessary for transferring thin tunnel $h$-BN and underlying single-layer graphene films. Finally, titanium and gold (5 nm/95 nm) electrodes are fabricated to electrically connect the top graphite and bottom graphene layers by using standard electron-beam lithography and lift-off procedures. All four high-purity (>99.995%) SC-TMD crystals were purchased from HQ Graphene with no additional dopants added during growth procedures.

**Electrical measurements.** All electrical measurements are carried out under a high vacuum condition below $10^{-5}$ Torr with a cryogen-free probe station and cryogen-free dilution refrigerator, whose base temperatures are 5.7 K and below 100 mK, respectively. We first characterize the electronic properties of our planar tunnel junctions with the probe station, and transfer the working devices to the dilution refrigerator for further in-depth measurements. With SC-mTMD films encapsulated by noninteracting high-quality $h$-BN and graphene, our devices are resilient to multiple thermal circulations. Current–voltage characteristics of the mTMD planar tunnel junctions are measured with a DC voltage source applied to the top graphite, and a current amplifier connected to the bottom graphene layer. Voltage output from the current amplifier is monitored through a digital multimeter, and differential conductance (d$I$/d$V_b$) is numerically obtained.

**Optical measurements.** Spectral information regarding the $A$-exciton peaks of the SC-mTMD films is measured with a micro-PL system pumped by a CW Nd: YAG laser at 532 nm. Pump light is vertically shone onto the SC-mTMD flakes through a ×50 objective lens, and the PL signals collected through the same lens are analyzed with a liquid nitrogen-cooled Si CCD detector with 550 -nm long pass filter. The spatial extent of the pump laser is confirmed to be 2 µm in diameter from knife-edge experiments. Energy splitting between the $A$- and $B$-exciton peaks of the SC-mTMDs is measured with a micro-reflectance spectroscopy setup equipped with a supercontinuum white light source. The diameter of the supercontinuum light source is estimated to be 4 µm. For temperature-dependent PL and reflectance measurements, our mTMD-based planar junctions are mounted in a liquid nitrogen-cooled cryostat.

**DFT for atomic defect states.** Density functional theory calculations are performed using the Vienna Ab initio Simulation Package (VASP)[39]. Ions are represented by projector-augmented wave (PAW) potentials[40], and generalized gradient approximation (PBE) is employed to describe the exchange-correlation functional[41]. A plane-wave basis set with an energy cutoff of 350 eV is employed to describe electronic wave functions. For defect-state calculations, we use an 8 × 8 supercell with consideration of the dipole, and the Γ point for Brillouin-zone integration is used for structural optimizations and total energy calculations. We find that the effects from the dipole correction are very small, with an energy

---

### Table 1 Experimentally identified material parameters of the four representative SC-mTMDs

Units: eV

|  | $E_g$ | $E_V$ | $E_C$ | $\Delta_{SO}$ | $\Delta_{K\text{-}\Gamma}$(VB) | $\Delta_{K\text{-}Q}$(CB) | $E_{opt}$ | $E_B$ |
|---|---|---|---|---|---|---|---|---|
| WS$_2$ | 2.88 | 1.95 | 0.93 | 0.38 | 0.26 | 0.12 | 2.051 | 0.83 |
| MoS$_2$ | 2.72 | 1.86 | 0.86 | 0.15 | 0.15 | 0.15 | 1.936 | 0.78 |
| WSe$_2$ | 2.56 | 1.21 | 1.35 | 0.43 | 0.86 | −0.15 | 1.724 | 0.82 |
| MoSe$_2$ | 2.46 | 1.59 | 0.87 | 0.20 | 0.54 | 0.13 | 1.638 | 0.82 |

All parameters listed are experimentally addressed by electron tunneling and optical spectroscopy measurements with SC-mTMD-based planar heterostructures. Energy-level assignments have uncertainty levels < ± 0.01 eV

difference of <1 meV for mMoS$_2$ (Supplementary Fig. 12). The calculated hexagonal lattice constants of mMoS$_2$, mMoSe$_2$, mWS$_2$, and mWSe$_2$ are 3.188 Å, 3.312 Å, 3.179 Å, and 3.325 Å, respectively. The atomic positions of all clusters are relaxed with residual forces smaller than 0.01 eV/Å.

## Data availability

All data supporting the findings of this study are available from the corresponding authors on request.

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

## Acknowledgements

This work was supported by a research grant for the Development of Converging Measurement Technology for Nanotechnology (KRISS-2018-GP2018–0019) funded by the Korea Research Institute of Standards and Science. This work was also supported by the Basic Science Research Program (NRF-2016R1A2B4008816 and NRF-2019R1A2C2004007) through the National Research Foundations of Korea.

## Author contributions

T.Y.J., H.K., and S.J. fabricated devices and performed electron tunneling spectroscopy measurements. T.Y.J. and K.J.Y. carried out optical spectroscopy measurements, S.-J.C. established an electron tunneling model of vertical 2D vdW heterostructures, and Y.-S.K. performed DFT calculations. High-quality *h*-BN crystals were synthesized by K.W. and T.T. T.Y.J., H.K., K.J.Y., S-J.C., Y.-S.K., and S.J. contributed in analyzing the data and preparing the paper.

## Additional information

**Competing interests:** The authors declare no competing interests.

