## [Transparent Peer Review File · Nature Communications]

Reviewers' comments:

Reviewer #1 (Remarks to the Author):

In the paper entitled "Atomic defect states and bandgap renormalization of semiconducting monolayer transition metal dichalcogenides assessed by high-fidelity electron tunneling and optical spectroscopy measurements", Tae Young Jeong and coworkers utilize electron-tunneling and optical measurements supplemented with DFT calculations to locate the mid-gap states originating from single-chalcogen vacancies in TMD monolayers. Due to the varying position of the defect states with respect to the band extrema in the different monolayers, the authors conclude that these defects are directly related to the charge dopings found in the experiments on TMDs – with MoS₂, WS₂, and MoSe₂ being n-type semiconductors and WSe₂ p-type.

The paper is well written, the figures are instructive, the experimental methods are state of the art and the results are really interesting. Yet, prior to publication I would like the authors to address some questions.

One main question I have is, how can the authors be sure that the in-gap states they see in their experiments are due to chalcogen vacancies? Also H, H₂, N, C, and Si can form these in-gap states, see Ref. 15. I think the measurements really show mid-gap states due to chalcogen vacancies but the authors should at least discuss the possibility of having other defects.

The second question concerns the theoretical calculations. The authors (just like others in many other publications before) do not include spin-orbit coupling in their calculations. Why? SOC will not only be important for the position of the conduction-band minimum at Q with respect to K, or the valence-band splitting, but it will also lead to a splitting of the defect states! I have seen this in my own calculations (yet unpublished) but it can also be seen in this paper:

Large spin-orbit splitting of deep in-gap defect states of engineered sulfur vacancies in monolayer WS₂

<https://arxiv.org/abs/1810.02896>

This shows that the "doublet" will split into 2 states. In fact, looking at Figs. 2(a) and 3(a) of the present manuscript I can also "clearly" see this splitting in their experiments (for WSe₂ better than WS₂). For the molybdenum-based TMD monolayers this splitting will be much smaller and thus might be undetectable. Additionally, including SOC might actually improve the agreement between experiments and theory for the position of the defect states with respect to the conduction- and valence-band edge (which is – looking at Fig. 4(c) – not that good at the moment).

Some minor details:

- The authors could compare their calculated position of the defect state with respect to the conduction-band minimum with other calculations as there are already so many publications on this topic. Also the formation energies presented in Figs. 4(a) and (b) have already been calculated – how do the results of the authors compare to published work?
- Did the authors include the dipole correction in their calculations?
- The measurements show that the conduction-band minimum at Q is 150 meV lower than the K minimum in WSe₂ which is a lot – nearly twice as large as in other publications, e.g.

Probing Critical Point Energies of Transition Metal Dichalcogenides: Surprising Indirect Gap of Single Layer WSe₂

<https://pubs.acs.org/doi/full/10.1021/acs.nanolett.5b01968>

This could be due to the interaction with the encapsulating h-BN and/or graphene. The authors should comment on this and also give some estimation how their results on the relative position of K with respect to Q or Γ with respect to K depend on this specific environment.

Reviewer #2 (Remarks to the Author):

Jeong et al. assessed the electron structure including mid-gap states originating from chalcogen atom vacancies, the quasiparticle energy gaps, exciton binding energies of mWS₂, mMoS₂, mWSe₂, and mMoSe₂ through electron tunneling spectroscopies, optical spectroscopies and DFT calculations. They also discussed the similarities and dissimilarities among these four SC-mTMDs regarding defects states. The topic of this work is interesting. Defects in 2D materials critically determine the electronic structure and other functionality of the materials. To precisely determine the defect states within the band structure is important. The tunneling and optical spectroscopy experiment in this work are carefully done and the data analyze is clear. The author claimed that they provide the most accurate and reliable materials parameters to date for the four materials they investigated. However, I believe the authors need to provide further evidence to really support their conclusion. So I suggest the manuscript to be rejected and can be reconsidered if further experimental information can be provided

1. The results obtained from both the tunneling and optical spectroscopy experiments are from a large area of the materials (much larger than the size of single S/Se vacancy or vacancy clusters). So the electronic structure characteristics the author got are actually average information over the whole materials. To correlate such information to a single defect type such as the S vacancy or Se vacancy as the author claim require more experimental evidence. The authors need to provide more characterization of the mWS₂, mMoS₂, mWSe₂, and mMoSe₂ they investigated. Is it really true that the dominate defect in these samples are S or Se vacancy? Are they sure that other defects such as Mo or W vacancy dose not exist in the sample they prepare?
2. The author just cite several reference to prove that the dominate defects in mWS₂, mMoS₂, mWSe₂, and mMoSe₂ vacancies. So in the discussion they just only focus on S or Se vacancy. Although the calculation results show that defect states induced by S and Se vacancy match the experiment, other type of defect also need to consider in their discussion, such as double vacancy, cation vacancy, etc.
3. Why only mWSe₂ need to fabricate a second type of mWSe₂ planar tunnel device with crystalline angles of top graphite and mWSe₂ layers tightly aligned in order to efficiently probe the electronic structures around the K point by minimizing tunnel-electron momentum mismatch, while mMoS₂, mMoSe₂ and mWS₂ are unnecessary? And the fabrication of the second type of mWSe₂ planar tunnel device should be described in details.
4. Page 18 "tunneling spectra related to the doublet e defect states in mWS₂ film remain as an isolated dI/dVb hump (Figure 2a), despite the fact that mWS₂ layers are expected to have a larger defect density of states than mMoSe₂, in which a much extended defect-induced dI/dVb bump is formed (Figure 3c)." Please explain this phenomenon.
5. The SC-mTMD films are encapsulated by non-interacting high-quality h-BN and graphene. Please explain the "non-interacting". As far as I know, there can be electric dipole between the interface of graphene and MoS₂. Can the author comment about the impact of interface states on their tunneling experiment results?
6. The dry transfer method that was used to transfer graphene to the pre-located h-BN flake on SiO₂/Si substrate should be described in details.

Reviewer #3 (Remarks to the Author):

Jeong and coworkers report a combination of tunneling and optical spectroscopy measurements with DFT calculations to identify the band structures and defect states of four monolayer semiconducting TMDs (MoS₂, MoSe₂, WS₂ and WSe₂). By comparing bulk tunneling spectroscopy from a graphite probe, the authors conclude that S and Se vacancies are prevalent defect sources in all four materials, and are responsible for in-gap defect states.

While I believe this study has elements of current interest, I have reservations about a number of the interpretations drawn from the data presented. For this reason, I believe that publication in any form would be premature at this time. Below, I list my most pressing concerns:

- 1) There are a number of ways in which I believe that features of the tunnel spectroscopy are potentially over-interpreted (Figs. 2 and 3). The dI/dV are averaged in some way to reveal “peaks” at low bias which are said to arise from defect states. The averaging procedure is a bit mysterious to me – for example, in Fig. 2a the noise looks fairly homogeneous from 0 to -0.5 V, yet a peak somehow emerges upon averaging. How can small remaining bumps in the data be identified unambiguously as arising from in-gap defect states? Even more concerning, the graphite/graphene tunneling response itself shows unexpected peaks in dI/dV in a number of samples (Fig. S1, in particular devices 3 and 4). How can the authors distinguish between low-bias peaks they observe arising to TMD defect states and those from non-ideal tunneling into the graphene substrate?
- 2) I’m not sure how the gap is being extracted in these samples. For all 4 materials, the dI/dV starts to diverge from the graphene background near the valence band edge at a much lower bias than the corresponding assignment of the band edge (the deviation regions are shaded in purple). Why is this not assigned to be the valence band edge, beyond a theoretical expectation that there should be defect states residing just above the valence band? In other words, how can the authors distinguish between those two cases experimentally?
- 3) Similarly, near the conduction band the authors use deviations from the expected graphite tunnel spectra to motivate broad regions of defect states in MoS₂ and MoSe₂. Taking for example the MoS₂ results in Fig. 2c, there appears to be reasonably good agreement with the graphite curve shown in the green dot-dashed line even at low bias (~-0.25 V). How can the authors be sure that the valence band edge is not at this bias instead of at -0.86 V?
- 4) Assigning the band edges to the lower biases where the dI/dV first diverges from the graphite background would instead yield smaller gaps. In such a case, the interpretation of in-gap defect states would then be opposite, with little evidence for their existence. The measured dI/dV signals in the gap are very small and are being compared to an already uncertain graphite/graphene background with arbitrary multiplicative factor, so it is difficult for me to believe interpretations based on small bumps or deviations from that “background,” especially those that appear only after the averaging procedure. Previous reports of the band gaps from the literature are typically smaller than the values reported here (see as just one example A. Raja et al., Nature Comm. 8, 15251 (2017)).
- 5) The authors argue that two different WSe₂ samples exhibit qualitatively different tunnel spectroscopy owing to different rotational alignment of the graphite and WSe₂. However, the Brillouin zones of graphite and WSe₂ are very different in size owing to the large lattice mismatch, hence it is not obvious to me that rotational alignment should actually matter that much in terms of the K-point momentum offsets and subsequent tunneling constant. The authors should present a more quantitative analysis if they want to make such a claim. Furthermore, why is this only an issue in WSe₂, and not in the results of the other three TMDs?
- 6) What is the experimental energy resolution for determining SOC splitting in the CB? All four TMDs are expected to exhibit some splitting, but a value is only reported here for WSe₂.
- 7) The figures are not referenced sequentially in the text.

Reviewer(s)' Remarks to the Author:

Reviewer #1

Comments:

In the paper entitled "Atomic defect states and bandgap renormalization of semiconducting monolayer transition metal dichalcogenides assessed by high-fidelity electron tunneling and optical spectroscopy measurements", Tae Young Jeong and coworkers utilize electron-tunneling and optical measurements supplemented with DFT calculations to locate the mid-gap states originating from single-chalcogen vacancies in TMD monolayers. Due to the varying position of the defect states with respect to the band extrema in the different monolayers, the authors conclude that these defects are directly related to the charge dopings found in the experiments on TMDs – with MoS₂, WS₂, and MoSe₂ being n-type semiconductors and WSe₂ p-type. The paper is well written, the figures are instructive, the experimental methods are state of the art and the results are really interesting. Yet, prior to publication I would like the authors to address some questions.

Response:

We thank the Reviewer for their high praise on the quality of our experiments and for finding our work interesting. Moreover, we appreciate their thoughtful comments and questions, which have made our manuscript better. Detailed point-by-point responses to each question with additional theoretical analyses are listed below, and all alterations to the manuscript are highlighted here in red. We hope our responses and revisions fully resolve the concerns that the Reviewer has raised.

1) One main question I have is, how can the authors be sure that the in-gap states they see in their experiments are due to chalcogen vacancies? Also H, H₂, N, C, and Si can form these in-gap states, see Ref. 15. I think the measurements really show mid-gap states due to chalcogen vacancies but the authors should at least discuss the possibility of having other defects.

Response:

We appreciate the Reviewer's important remarks on the origins of the mid-gap states in our experiments. As pointed out by the Reviewer, it is possible that impurities relating to H, H₂, N, C, and Si adatoms along with chalcogen atom vacancies form the mid-gap defect states in monolayer SC-TMDs. Although it would require additional full theoretical and experimental attentions to address all possible defect states from the above-mentioned impurities in detail, we are convinced that the spectral features observed in our experiments should be from the S- and Se-atom vacancies (V_S and V_{Se}) in SC-TMD monolayer films, as conferred below.

At first, it is not plausible that H and H₂ adatoms would stay on the monolayer SC-TMDs after several steps of the device fabrication procedures, including a high-temperature annealing at 250 °C. In our calculation, the binding energies of H on MoS₂, MoSe₂, WS₂, and WSe₂ films are found to be negative with respect to free H₂ molecules: -1.9, -2.1, -2.2, and -2.4 eV/H, respectively, suggesting that H atoms are likely to desorb from the TMDs and form H₂ molecules. The calculated binding energies of H₂ (physisorption) on the surface of TMD films are a few tens of meV, which allow H₂ molecules to be easily desorbed from the SC-TMD films. Next, Ma *et al.* and Haldar *et al.* also reported that the formation energy of N₂ chemisorption to monolayer TMDs is negative, which makes N₂ as well highly unlikely to be the origin of the observed mid-gap states [D. Ma *et al.* Phys. Chem. Chem. Phys. **19**, 26022 (2017); S. Haldar *et al.* Phys. Rev. B **92**, 235408, (2015)]. The mid-gap states from C- and Si-induced defects could be rare in our monolayer TMDs, when compared with the abundant chalcogen-atom vacancies.

There exist a few articles on the defect states from transition-metal atom vacancies and substitutions that could be formed either through external dopings or in reorganized crystals during growth process [J. Gao *et al.* Adv. Mat. **28**, 44 (2016); J. Suh *et al.* Nano Lett. **14**, 6976 (2014)]. However, much higher formation energies of transition-metal atom vacancies (V_{Mo} and V_W), as reported by W. Zhou *et al.*, D. Ma *et al.*, and S. Haldar *et al.* [W. Zhou *et al.* Nano Lett. **13**, 2615 (2013); D. Ma *et al.* Phys. Chem. Chem. Phys. **19**, 26022 (2017); S. Haldar *et al.* Phys. Rev. B **2**, 235408, (2015)] suggest that the missing transition-metal atoms are not energetically favorable to chalcogen-atom vacancies with far lower formation energies.

Although we concur that it is still possible that there may exist several other defects in our monolayer SC-TMDs other than the missing chalcogen atoms, experimental identifications of such defect states are daunting, especially in measurement schemes with planar tunnel junctions whose active areas are as large as several microns in width and length. In our devices, observed mid-gap states are ensembles of all the available defect states, whose spectral features could be easily overshadowed by the more dominant defect sources: chalcogen-atom vacancies in Mo- and W-based monolayer SC-TMD films.

Fully agreeing with the Reviewer's suggestion, we have added the following argument in the main text on the possibility of other defect sources in our devices, with new references cited.

Added in the main text (page 19):

“... Such consistent energy-level shifting of a_1 and e defect states to higher energies (toward conduction bands) presents a strong experimental implication that chalcogen-atom vacancies in SC-mTMDs can work as dopants, especially as dominant donor states in mWS₂, mMoS₂, and mMoSe₂ films.

Here, it is worth mentioning that it is possible there exist several other defects in the monolayer SC-TMDs such as H₂ or N₂ adatoms, transition metal-atom vacancies, or others. However, experimental identifications of such scant defect states are daunting, especially in our planar tunnel junctions in which active areas are as large as several microns in width and length. In our devices, the observed mid-gap states are ensembles of all the available defect states, whose spectral features could be easily overshadowed by the more abundant defect sources;

missing chalcogen atoms in Mo- and W-based monolayer SC-TMD films. Moreover, the formation energies of chalcogen-atom vacancies are favorable to many other defect sources. For example, much higher formation energies of transition-metal atom vacancies (V_{Mo} and V_{W})^{17,52–54} suggest that the missing transition-metal atoms are not energetically favorable to chalcogen-atom vacancies with far lower formation energies, not to mention that the spectral locations of such defect states are different in energy from those of chalcogen-atom vacancies (Supplementary Figure 11). In addition, we find that the binding energies of H on SC-TMD films are negative with respect to free H₂ molecules; -1.9, -2.1, -2.2 and -2.4 eV/H, respectively for MoS₂, MoSe₂, WS₂ and WSe₂, as similar to N₂ chemisorption,^{52,53} suggesting that H atoms are more likely to desorb from the TMDs and form H₂ molecules instead. The calculated binding energies of H₂ on the surface of TMD films are a few tens of meV, which allow H₂ to be easily detached from the SC-TMD films as well.”

2) The second question concerns the theoretical calculations. The authors (just like others in many other publications before) do not include spin-orbit coupling in their calculations. Why? SOC will not only be important for the position of the conduction-band minimum at Q with respect to K, or the valence-band splitting, but it will also lead to a splitting of the defect states! I have seen this in my own calculations (yet unpublished) but it can also be seen in this paper:

Large spin-orbit splitting of deep in-gap defect states of engineered sulfur vacancies in monolayer WS₂

<https://arxiv.org/abs/1810.02896>

This shows that the "doublet" will split into 2 states. In fact, looking at Figs. 2(a) and 3(a) of the present manuscript I can also "clearly" see this splitting in their experiments (for WSe₂ better than WS₂). For the molybdenum-based TMD monolayers this splitting will be much smaller and thus might be undetectable. Additionally, including SOC might actually improve the agreement between experiments and theory for the position of the defect states with respect to the conduction- and valence-band edge (which is – looking at Fig. 4(c) – not that good at the moment).

Response:

We thank the Reviewer for their important remarks on spin-orbit coupling (SOC) in TMD monolayers and its crucial role not only for the intrinsic electronic structures but also for the defect states. We admit that we did not consider SOC in our initial calculation for the sake of simplicity. In this revision, however, we have included the SOC effects in our updated DFT calculations and find out that the doublet *e* defect states indeed become split in the W-based TMD monolayers, as marked with red arrows in Fig. R1 (c) for WS₂ and R1 (d) for WSe₂. As consistent with the previous report by B. Schuler *et al.* [B. Schuler *et al.* <https://arxiv.org/abs/1810.02896>] and the Reviewer’s remark, SOC-induced defect-state

splittings are confirmed to be minimal in Mo-based TMD monolayers in our DFT calculations as well; see Fig. R1 (a) for MoS₂ and Fig. R1 (b) for MoSe₂.

As the Reviewer pointed out, a pair of distinctive dI/dV_b peaks as marked with orange arrows in Fig. 3a for monolayer WSe₂ and some mid-gap states in Fig. 2a for mWS₂ could be related to the SOC-induced defect-state splittings: this observation could serve as another piece of decisive evidence that chalcogen-atom vacancies are the dominant defect sources in our SC-TMD monolayers. However, we should be careful in labeling the SOC effect as the leading source for these isolated dI/dV_b peaks in Fig. 3a for mWSe₂. As pointed out in our original manuscript, the set of dI/dV_b peaks in mWSe₂ (Fig. 3a) along with the amplified and extended dI/dV_b in mMoS₂ (Fig. 2c) and mMoSe₂ (Fig. 3c) can also be attributed to double chalcogen vacancies, vacancy chains or clusters.

Figure R1 (a–d) Calculated defect DOS for four SC-mTMD films without (upper row) and with (lower row) consideration of SOC effects.

In this revised manuscript, we now include the defect density of states (DOS) of double chalcogen vacancies ($V_S V_S$ and $V_{Se} V_{Se}$) for all four SC-mTMDs (Figure R2), whose spectral locations are close to those of single chalcogen vacancies with additionally extended DOS features, which are in better agreement with experimental observations of mMoS₂ and mMoSe₂ (See Fig. 4a). When it comes to mWSe₂, the spectral positions for the observed dI/dV_b peaks in Fig. 3a are close to the doublet e defect states from both single Se vacancies with SOC (Fig. R1 (d)) and double Se vacancies (Fig. R2 (d)). Thus, at this moment, we are afraid to admit that it is not certain whether the SOC or double Se vacancies or both are the sources of the isolated set of dI/dV_b bumps in mWSe₂ (Fig. 3a), which calls for further theoretical and experimental works.

To note, the upper row in Fig. R2 indicates the DOS from the vacancies of transition metal atoms (V_{Mo} and V_W) for all four SC-TMD layers, and the spectral locations for such defects are not consistent with experimental data, reinforcing that chalcogen-atom vacancies are indeed the dominant defect sources in our SC-TMD monolayers.

Figure R2 (a–d) Calculated defect DOS for four SC-mTMD films from the missing of transition-metal atoms (upper row) and double chalcogen-atom vacancies (lower row).

To include these crucial findings, we have added the following arguments in the main text with a new reference cited, and modified Fig. 4a with updated DFT calculations with the considerations of SOC (Fig. R1) and double chalcogen vacancies (Fig. R2). In addition, we have added Figures R1 and R2 in the Supplementary Information.

Added in the main text (page 17):

“We numerically calculate the atomic defect density of states (DOS) due to chalcogen-atom vacancies in the four respective SC-mTMDs with a consideration of the SOC effect, and present overlaid plots with experimental data in Figure 4a. In these calculations, we do not consider Coulombic many-body effects since a rather simplified DFT is sufficient to capture the varying spectral locations of the a_1 and e defect states in SC-mTMDs. It is crucial to point out that the SOC effect in SC-mTMDs influences not only the intrinsic electronic structures but also the defect states from chalcogen-atom vacancies.⁵¹ As displayed in Figure 4a, the SOC effect splits the doublet e defect states as large as ≈ 0.2 eV in mWS₂ and mWSe₂, while the splitting is minimal in mMoS₂ and mMoSe₂ (Supplementary Figure 10). As experimentally and theoretically confirmed, the a_1 defect states of the Mo-based films extend further into the energy gap, while the majority of the a_1 states of the W-based films hide under the valence bands. Spectral locations of the doublet (e) defect states in mWS₂ and mWSe₂ films are close to the theoretically expected positions, and calculated e defect states in mMoS₂ and mMoSe₂ films are in the vicinity of the expanded defect-induced dI/dV_b . We calculate the defect DOS induced by double chalcogen-atom vacancies ($V_S V_S$ or $V_{Se} V_{Se}$) and present them with dotted grey lines in Figure 4a. As noted previously, it is confirmed that the e - e divacancy defects in SC-mTMD layers become hybridized and form additional mid-gap states inside the energy gap, further extended toward the conduction-band edge. We consider charge neutral chalcogen-atom vacancies in our DFT simulations, and find that the calculated spectral locations of both a_1 and e

defect states in mMoS₂, mMoSe₂, and mWS₂ films are consistently shifted to higher energies than the experimentally identified locations ...”

Added in the main text (page 20):

“Finally, a few dI/dV_b humps for the doublet defect states in the mWSe₂ film (Figure 3a) could imply that some selenium-atom vacancies become hybridized, justified by a relatively high vacancy-binding energy ($E_{\text{bind}}(\text{mWSe}_2) = 0.086 \text{ eV}$), although the overall number of selenium-atom vacancies is expected to be the lowest among the four SC-mTMDs. However, we should note that these dI/dV_b bumps could be attributed to the SOC-induced defect-state splittings (Figure 4a) as well, and not just the double Se vacancies. At this moment, we cannot say with certainty whether the SOC or divacancy defects or both are the sources for the isolated set of dI/dV_b bumps in mWSe₂ (Figure 3a), which calls for further theoretical and experimental works.”

Some minor details:

3) The authors could compare their calculated position of the defect state with respect to the conduction-band minimum with other calculations as there are already so many publications on this topic. Also the formation energies presented in Figs. 4(a) and (b) have already been calculated – how do the results of the authors compare to published work?

Response:

We have found that our calculations (Kohn–Sham) on the chalcogen-atom vacancies are consistent with other published works. Among them, Komsa *et al.* [H.-P. Komsa and A. V. Krasheninnikov, Phys. Rev. B **91**, 125304 (2015)] reported that the doublet e defect state of mMoS₂ is positioned at $\approx E_c - 0.5 \text{ eV}$, which is close to our calculation $\approx E_c - 0.6 \text{ eV}$ in this report, and the value from a previous report by one of us [Noh. *et al.*, Phys. Rev. B **89**, 205417 (2014)]. In addition, Mahjouri-Samani *et al.* [M. Mahjouri-Samani *et al.* Nano Lett. **16**, 5213-5220 (2016)] reported e defect states of mMoSe₂ from Se vacancies are formed at $\approx E_c - 0.5 \text{ eV}$, and the position is also matched to our calculation, $\approx E_c - 0.5 \text{ eV}$.

In our calculation (re-labeled as Fig. 4b), the formation energy of sulfur vacancy in mMoS₂ is $\approx 1.3 \text{ eV}$ in the Mo-rich limit condition, and the energy is close to the values (1.2–1.5 eV) in several other reports [Noh. *et al.*, Phys. Rev. B **89**, 205417 (2014); Zhou *et al.*, Nano Lett. **13**, 2615 (2013); Komsa *et al.*, Phys. Rev. B **91**, 125304 (2015); Haldar *et al.*, Phys. Rev. B **92**, 235408 (2015)]. The formation energy of selenium vacancies in mMoSe₂ is estimated to be $\approx 1.5 \text{ eV}$ in our calculation and the value is also consistent with the value of 1.5 eV, reported by Haldar *et al.* [Haldar *et al.*, Phys. Rev. B **92**, 235408 (2015)]. In addition, the formation energy of sulfur vacancy in mWS₂ is found here to be $\approx 1.4 \text{ eV}$, with Haldar *et al.* reporting a similar

value, ≈ 1.5 eV [Haldar *et al.*, Phys. Rev. B **92**, 235408 (2015)]. Last, the formation energy of selenium vacancy in mWSe₂ films is ≈ 1.7 eV in our calculation, which is close to other reported formation energies of 2 eV [Zhang *et al.*, Phys. Rev. Lett. **119**, 046101 (2017)] and 1.9 eV [Haldar *et al.*, Phys. Rev. B **92**, 235408 (2015)]. Haldar *et al.* [Haldar *et al.*, Phys. Rev. B **92**, 235408 (2015)] systemically showed that the formation energy of anion vacancies increases in the orders of XS₂, XSe₂, and XTe₂ (X=Mo, W) in type-VI TMD materials, and the formation energies in W-based TMDs are higher than those in M-based TMDs. All of this are consistent with our results in Fig. 4b.

4) Did the authors include the dipole correction in their calculations?

Response:

Yes! We included the dipole correction in our initial calculations. Actually, the effects from dipole correction are very small with our large-size supercell (8×8) and vacuum thickness of 2 nm. Figures R3 (a) and (b) show the plane-averaged local Hartree potentials along the perpendicular direction to two-dimensional layers, with (a) and without (b) dipole correction for sulfur vacancies in monolayer MoS₂ films. The calculated total energies are almost identical for both cases: the difference is less than 1 meV.

Figure R3 (a,b) Plane-averaged local Hartree potentials for single sulfur vacancies in mMoS₂ films with (a) and without (b) dipole corrections.

To clarify this point, we have added the following discussion on dipole correction in the Methods section, and Figure R3 in the Supplementary Information.

Added in the Methods (page 33):

DFT for atomic defect states

“... For defect-state calculations, we use an 8×8 supercell **with consideration of the dipole**, and the Γ point for Brillouin-zone integration is used for structural optimizations and total energy calculations. **We find that the effects from the dipole correction are very small, with an energy difference of less than 1 meV for mMoS₂ (Supplementary Figure 12).** The calculated hexagonal lattice constants of mMoS₂, mMoSe₂, mWS₂, and mWSe₂ are 3.188 Å, 3.312 Å, 3.179 Å, and 3.325 Å, respectively. ...”

5) The measurements show that the conduction-band minimum at Q is 150 meV lower than the K minimum in WSe₂ which is a lot – nearly twice as large as in other publications, e.g. “Probing Critical Point Energies of Transition Metal Dichalcogenides: Surprising Indirect Gap of Single Layer WSe₂”, <https://pubs.acs.org/doi/full/10.1021/acs.nanolett.5b01968>

This could be due to the interaction with the encapsulating h -BN and/or graphene. The authors should comment on this and also give some estimation how their results on the relative position of K with respect to Q or Γ with respect to K depend on this specific environment.

Response:

We thank the Reviewer for their inquiries on the detailed electronic structures of monolayer TMD films. We are aware of the article by C. Zhang *et al.* [Nano Lett. **15**, 6494-6500 (2015)] and refer to it as reference #27 in the manuscript while discussing the indirect energy gap of monolayer WSe₂. In that report, C. Zhang *et al.* reported the conduction-band minimum at the Q point is roughly 80 meV lower than the K -point minimum, which is about a half of our observation \approx 150 meV.

It is well recognized that the electronic structures of SC-TMDs around the conduction band edges, thus the detailed locations of the K - and Q -point minima, are sensitive to the p -orbital structures of chalcogen (S and Se) atoms. Thus, the spacing between K - and Q -points in WSe₂ monolayer films becomes dependent on (i) how the films are prepared, i.e. CVD grown (Ref. #27) on graphite *vs.* mechanically isolated from single crystals and transferred on graphene/ h -BN heterostacks (our work), and (ii) how the monolayer films are incorporated with other device structures, i.e. directly exposed to vacuum and a scanning probe (Ref. #27) *vs.* encapsulated with tunnel h -BN and a graphite probe (our work). As the Reviewer suggested, we concur that such

structural differences ultimately lead to the K-Q spacing variations in the monolayer WSe₂ observed from the scanning tunneling measurements (Ref. #27) and our planar tunnel junctions.

Following the Reviewer's suggestion, we have added the following comments in the main text.

Added in the main text (page 16):

“... With these assignments, we assert that mWSe₂ is an indirect band-gap and weakly doped *p*-type semiconductor, whose quasiparticle energy gap is as large as $E_g = 2.56$ eV ($E_C = 1.35$ eV and $E_V = 1.21$ eV) and exciton binding energy as high as $E_B = 0.82$ eV ($E_{opt} = 1.724$ eV). Furthermore, SOC-split conduction bands at the K point of the mWSe₂ film are found to be $\Delta_{SO,C}$ (mWSe₂) = 0.03 eV. Recently, Wang *et al.* reported the **similar** value ($\Delta_{SO,C}$ (mWSe₂) \approx 0.04 eV) by optical reflectance and PL spectroscopy measurements.⁵⁰ **Here, it is worth pointing out that the notable difference at the K-Q conduction band minima between our measurement (\approx 150 meV) and the STM measurement with mWSe₂ grown on graphite films by chemical vapor deposition (\approx 80 meV)²⁷ could be attributed to the Se *p*-orbital structures being modified through the various sample-fabrication procedures and device geometries.** Complete material property assignments for mWSe₂ are illustrated in Figure 3b.”

Reviewer #2

Comments:

Jeong et al. assessed the electron structure including mid-gap states originating from chalcogen atom vacancies, the quasiparticle energy gaps, exciton binding energies of mWS_2 , $mMoS_2$, $mWSe_2$, and $mMoSe_2$ through electron tunneling spectroscopies, optical spectroscopies and DFT calculations. They also discussed the similarities and dissimilarities among these four SC-mTMDs regarding defects states. The topic of this work is interesting. Defects in 2D materials critically determine the electronic structure and other functionality of the materials. To precisely determine the defect states within the band structure is important. The tunneling and optical spectroscopy experiment in this work are carefully done and the data analyze is clear. The author claimed that they provide the most accurate and reliable materials parameters to date for the four materials they investigated. However, I believe the authors need to provide further evidence to really support their conclusion. So I suggest the manuscript to be rejected and can be reconsidered if further experimental information can be provided.

Response:

We thank the Reviewer for their appreciation on the quality of our works and for agreeing with us on the research motivation: to accurately assess the electronic structures of SC-TMD materials, and understand that the roles of common atomic defects in such materials are of critical importance to fully utilize their high functionalities in both scientific and engineering applications. Moreover, we appreciate the Reviewer's efforts in reading our manuscript thoroughly, and for raising genuine concerns on the origins of defect states, hoping to make our manuscript better. With additional theoretical analyses, here we list detailed point-by-point responses to each question. All the alterations made in the manuscript are highlighted here in red. We hope our revisions fully resolve the concerns raised.

1) The results obtained from both the tunneling and optical spectroscopy experiments are from a large area of the materials (much larger than the size of single S/Se vacancy or vacancy clusters). So the electronic structure characteristics the author got are actually average information over the whole materials. To correlate such information to a single defect type such as the S vacancy or Se vacancy as the author claim require more experimental evidence. The authors need to provide more characterization of the mWS_2 , $mMoS_2$, $mWSe_2$, and $mMoSe_2$ they investigated. Is it really true that the dominate defect in these samples are S or Se vacancy? Are they sure that other defects such as Mo or W vacancy dose not exist in the sample they prepare?

Response:

Yes! As the Reviewer points out, the observed spectra in our planar tunnel junctions are the ensembles of all possible defect states since the active junction areas are as large as several microns in width and length, differing from atomic-scale investigations by STM and TEM. Indeed, identifying the origins of the defect sources in our SC-TMD monolayers is the first and foremost assignment to analyze our experimental observations. It is possible that there exist several other atomic defect states from varying sources other than the chalcogen atom vacancies in our monolayer SC-TMDs. However, as we explain while answering the similar inquiry raised by the first Reviewer and the questions below, we are convinced that the most dominant and prevalent defects in SC-mTMD films are chalcogen-atom vacancies (V_S and V_{Se}), rather than missing transition-metal atoms (V_{Mo} and V_W) or adatoms of H_2 , N, Si, or C.

In short, the much higher defect formation energies of Mo- and W-atom vacancies suggest that missing transition-metal atoms, with their far lower formation energies, are not energetically favorable to chalcogen atom vacancies. Moreover, as shown in Figure R4 below, the calculated defect densities of states (DOS) from the vacancies of transition-metal atoms are not consistent with our observations. Additionally, our calculations suggest that H and H_2 adatoms would not stay on the monolayer SC-TMDs because of their either negative or small binding energies. In our calculation, the binding energies of H on MoS_2 , $MoSe_2$, WS_2 , and WSe_2 films are found to be negative with respect to free H_2 molecules: -1.9 , -2.1 , -2.2 and -2.4 eV/H, respectively, suggesting that H atoms are likely to desorb from the TMDs and form H_2 molecules. The calculated binding energies of H_2 (physisorption) on the surface of TMD films are a few tens of meV, which allows H_2 to be easily detached from the SC-TMD films. Ma *et al.* and Haldar *et al.* reported that the formation energy of N_2 chemisorption to monolayer TMDs becomes negative, suggesting that N_2 adatoms are highly unlikely to be the source of the observed mid-gap states either [D. Ma *et al.* Phys. Chem. Chem. Phys. **19**, 26022 (2017), S. Haldar *et al.* Phys. Rev. B **92**, 235408, (2015)].

Although we concur that it is highly plausible that there exist several defect states in our monolayer SC-TMDs other than the missing of chalcogen atoms, experimental identifications of such defect states are also daunting, especially in measurement schemes with planar tunnel junctions. As stated before, the observed mid-gap states in our devices are ensembles of all available defect states, whose spectral features could be overshadowed with ease by the dominant defects: chalcogen-atom vacancies in Mo- and W-based monolayer SC-TMD films.

2) The author just cite several reference to prove that the dominate defects in mWS_2 , $mMoS_2$, $mWSe_2$, and $mMoSe_2$ vacancies. So in the discussion they just only focus on S or Se vacancy. Although the calculation results show that defect states induced by S and Se vacancy match the experiment, other type of defect also need to consider in their discussion, such as double vacancy, cation vacancy, etc.

Response:

As explained in the previous response, the mid-gap states observed in our planar tunnel junctions are due to the most prevalent defect sources in monolayer SC-TMD films. We note that the defect densities of various atomic defects are directly related to their relative defect-formation energies. In a previous report by one of the current authors, Dr. Y-S. Kim [Noh. *et al.*, Phys. Rev. B **89**, 205417 (2014)], it was shown that the formation energies of cation vacancies (V_{Mo} and V_W) are higher than those of single chalcogen-atom vacancies, proving that missing Mo- and W-atoms in SC-mTMDs are sparser than V_S and V_{Se} . Moreover, as indicated with the blue curves in Figure R4 below, the calculated defect DOS from the cation vacancies are positioned below the mid-gap point, close to the valance-band edges, all of which are not consistent with our experimental results.

Figure R4 (a–d) Calculated defect DOS for four SC-mTMD films from the missing of transition-metal atoms (upper row) and double chalcogen-atom vacancies (lower row).

As for the double chalcogen vacancies ($V_S V_S$ and $V_{Se} V_{Se}$), we have pointed out in the manuscript that the amplified and extended dI/dV_b structures observed in mMoS₂ (Fig. 2c) and mMoSe₂ (Fig. 3c) are attributed to double chalcogen vacancies and vacancy clusters. Although we cited several references to support this claim, including an article coauthored by one of us [Reference #44, Vancso *et al.* Sci. Rep. **6**, 29726 (2016)], we admit that the original version was not clear enough to explain the impacts of double chalcogen vacancies on the mid-gap states of monolayer SC-TMD films. As shown in Figure R4 and the revised Figure 4a in the main text, we calculate defect DOS from double chalcogen vacancies for all four SC-mTMD films, and the spectral positions are closely matched to the experimental observations for mMoS₂ and mMoSe₂. Based on these, we are certain that chalcogen-atom vacancies are the dominant defect sources in SC-mTMDs, and double chalcogen vacancies prevail in mMoS₂ and mMoSe₂.

When it comes to mWSe₂, it is likely that double Se vacancies could be the main cause of the distinctive pair of dI/dV_b bumps inside the energy gap, as marked with orange arrows in Fig. 3a. However, we need to point out that those dI/dV_b humps could also be induced from spin-orbit coupling (SOC) effects. As the first Reviewer indicated and our new calculations revealed

(Figure R1 and the revised Fig. 4a), the doublet e defect states, especially in W-based mTMD films, become split. The splitting becomes as large as 0.2 eV with SOC effects considered (Fig. R1 (d)), and the spectral positions are closely matched to the mid-gap state positions from double Se vacancies (Fig. R4 (d)). In comparison, the SOC-induced defect-state splitting is found to be minimal in Mo-based monolayer TMDs—see Fig. R1 (a) for mMoS₂ and Fig. R1 (b) for mMoSe₂. Thus, at this moment, we are afraid to say that it is not certain whether SOC or double Se vacancies or both are the leading sources for the dI/dV_b bumps, and thus more theoretical and experimental works should be conducted to clarify the origins of the doublet e defect states in mWSe₂ (Fig. 3a).

To better clarify the defect-state origins, we have added the following arguments in the main text with new references cited, and modified Figure 4a with updated DFT calculations with SOC and double chalcogen-atom vacancy considered. In addition, we have added Figure R1 and Figure R4 in the Supplementary Information.

Added in the main text (page 17):

“We numerically calculate the atomic defect density of states (DOS) due to chalcogen-atom vacancies in the four respective SC-mTMDs with a consideration of the SOC effect, and present overlaid plots with experimental data in Figure 4a. In these calculations, we do not consider Coulombic many-body effects since a rather simplified DFT is sufficient to capture the varying spectral locations of the a_1 and e defect states in SC-mTMDs. It is crucial to point out that the SOC effect in SC-mTMDs influences not only the intrinsic electronic structures but also the defect states from chalcogen-atom vacancies.⁵¹ As displayed in Figure 4a, the SOC effect splits the doublet e defect states as large as ≈ 0.2 eV in mWS₂ and mWSe₂, while the splitting is minimal in mMoS₂ and mMoSe₂ (Supplementary Figure 10). As experimentally and theoretically confirmed, the a_1 defect states of the Mo-based films extend further into the energy gap, while the majority of the a_1 states of the W-based films hide under the valence bands. Spectral locations of the doublet (e) defect states in mWS₂ and mWSe₂ films are close to the theoretically expected positions, and calculated e defect states in mMoS₂ and mMoSe₂ films are in the vicinity of the expanded defect-induced dI/dV_b . We calculate the defect DOS induced by double chalcogen-atom vacancies ($V_S V_S$ or $V_{Se} V_{Se}$) and present them with dotted grey lines in Figure 4a. As noted previously, it is confirmed that the e - e divacancy defects in SC-mTMD layers become hybridized and form additional mid-gap states inside the energy gap, further extended toward the conduction-band edge. We consider charge neutral chalcogen-atom vacancies in our DFT simulations, and find that the calculated spectral locations of both a_1 and e defect states in mMoS₂, mMoSe₂, and mWS₂ films are consistently shifted to higher energies than the experimentally identified locations ...”

Added in the main text (page 19):

“... Such consistent energy-level shifting of a_1 and e defect states to higher energies (toward conduction bands) presents a strong experimental implication that chalcogen-atom vacancies in

SC-mTMDs can work as dopants, especially as dominant donor states in mWS₂, mMoS₂, and mMoSe₂ films.

Here, it is worth mentioning that it is possible there exist several other defects in the monolayer SC-TMDs such as H₂ or N₂ adatoms, transition metal-atom vacancies, or others. However, experimental identifications of such scant defect states are daunting, especially in our planar tunnel junctions in which active areas are as large as several microns in width and length. In our devices, the observed mid-gap states are ensembles of all the available defect states, whose spectral features could be easily overshadowed by the more abundant defect sources; missing chalcogen atoms in Mo- and W-based monolayer SC-TMD films. Moreover, the formation energies of chalcogen-atom vacancies are favorable to many other defect sources. For example, much higher formation energies of transition-metal atom vacancies (V_{Mo} and V_{W})^{17,52-54} suggest that the missing transition-metal atoms are not energetically favorable to chalcogen-atom vacancies with far lower formation energies, not to mention that the spectral locations of such defect states are different in energy from those of chalcogen-atom vacancies (Supplementary Figure 11). In addition, we find that the binding energies of H on SC-TMD films are negative with respect to free H₂ molecules; -1.9, -2.1, -2.2 and -2.4 eV/H, respectively for MoS₂, MoSe₂, WS₂ and WSe₂, as similar to N₂ chemisorption,^{52,53} suggesting that H atoms are more likely to desorb from the TMDs and form H₂ molecules instead. The calculated binding energies of H₂ on the surface of TMD films are a few tens of meV, which allow H₂ to be easily detached from the SC-TMD films as well.”

Added in the main text (page 20):

“Finally, a few dI/dV_b humps for the doublet defect states in the mWSe₂ film (Figure 3a) could imply that some selenium-atom vacancies become hybridized, justified by a relatively high vacancy-binding energy ($E_{\text{bind}}(\text{mWSe}_2) = 0.086 \text{ eV}$), although the overall number of selenium-atom vacancies is expected to be the lowest among the four SC-mTMDs. However, we should note that these dI/dV_b bumps could be attributed to the SOC-induced defect-state splittings (Figure 4a) as well, and not just the double Se vacancies. At this moment, we cannot say with certainty whether the SOC or divacancy defects or both are the sources for the isolated set of dI/dV_b bumps in mWSe₂ (Figure 3a), which calls for further theoretical and experimental works.”

3) Why only mWSe2 need to fabricate a second type of mWSe2 planar tunnel device with crystalline angles of top graphite and mWSe2 layers tightly aligned in order to efficiently probe the electronic structures around the K point by minimizing tunnel-electron momentum mismatch, while mMoS2, mMoSe2 and mWS2 are unnecessary? And the fabrication of the second type of mWSe2 planar tunnel device should be described in details

Response:

We appreciate the Reviewer’s careful reading of our manuscript and genuine inquires on our device schemes. As we discussed in the manuscript, we are certain that planar tunnel junctions with the crystalline angles of top graphite and mTMD layers aligned should make it efficient to probe the electronic structures of SC-mTMD films, especially around the K point. While analyzing our data, however, it is not necessary to rely on the second-type tunnel devices for mWS₂, mMoS₂, and mMoSe₂, because we can identify all the important characteristic features that are discussed in the current study. For the mWSe₂ device, however, dI/dV_b spectra from the first device were not clear enough to identify the electronic structures around the valence- and conduction-band edges, requiring us to prepare the second-type mWSe₂ planar junction hoping to observe the tunnel dI/dV_b features not revealed in the first device.

Figure R5 displays optical images of all the SC-mTMD-based planar tunnel devices discussed in the manuscript, with estimated twist angles of SC-mTMD films and underlying single-layer graphene with respect to top graphite probes given in each panel and summarized in the lower-right table. As marked in Fig. R5 (e), the crystalline directions of the top graphite probe and the mWSe₂ film are indeed closely aligned at $\theta_{WSe_2} \approx 0^\circ$ for the second-type device.

Figure R5 Optical images and twist angles of SC-mTMD films and underlying graphene with respect to the top graphite for (a) mWS₂, (b) mMoS₂, (c) mMoSe₂, and mWSe₂ devices without (d) and with (e) aligned angles. Scale bars are 20 μ m.

Following the Reviewer's suggestion, we have added extra explanation on the fabrication of the second-type devices in the Methods section in the revised manuscript, as found in our response to 6) below, and have also added Figure R5 with table in the Supporting Information.

4) Page 18 “tunneling spectra related to the doublet e defect states in mWS_2 film remain as an isolated dI/dV_b hump (Figure 2a), despite the fact that mWS_2 layers are expected to have a larger defect density of states than $mMoSe_2$, in which a much extended defect-induced dI/dV_b bump is formed (Figure 3c).” Please explain this phenomenon.

Response:

As elaborated above while answering the Reviewer's first and second inquiries, it is discussed that double chalcogen vacancies, vacancy clusters or chains generate doublet e defect states with much wider spectral locations inside the energy gap when compared with the defect states from individual chalcogen-atom vacancies. For example, the calculated defect DOSs due to the double chalcogen vacancies in our four SC-mTMD films (Fig. R4) well match to the experimentally identified doublet e defect states, with the agreement essentially outstanding for Mo-based SC-mTMDs. Moreover, the vacancy-binding energy of $mMoSe_2$ ($E_{\text{bind}}(mMoSe_2) = 0.126$ eV) is the highest among the four SC-mTMDs, which supports that double Se vacancies or vacancy chains can be formed with relative ease. In comparison, the vacancy-binding energy of mWS_2 ($E_{\text{bind}}(mWS_2) = 0.033$ eV) is the lowest, providing a compelling argument that sulfur vacancies in mWS_2 films tend to remain as isolated defects, and our experimental data prove as such. For $mWSe_2$, double Se vacancies could be the cause of the pair of isolated dI/dV_b bumps, since the vacancy-binding energy of $mWSe_2$ ($E_{\text{bind}}(mWSe_2) = 0.086$ eV) is reasonably high, almost three times larger than the E_{bind} of mWS_2 films. As discussed above, however, those distinctive dI/dV_b peaks could be also induced from the SOC effect. Figure R1 (d) shows that the SOC makes doublet e defect states in W-based SC-mTMD films become split, as large as 0.2 eV, and these spectral locations happen to be aligned with the mid-gap state from double Se vacancies.

5) The SC-mTMD films are encapsulated by non-interacting high-quality h -BN and graphene. Please explain the “non-interacting”. As far as I know, there can be electric dipole between the interface of graphene and MoS_2 . Can the author comment about the impact of interface states on their tunneling experiment results?

Response:

What we implied by claiming “non-interacting” h -BN and graphene layers in 2D van der Waals (vdW) heterostructures was a reference to either strong chemical bonding or orbital overlaps; therefore, interface states are not introduced across the vdW gaps in the heterostructures of h -BN,

graphene, and SC-mTMD films. In terms of the dipole, yes! There exist interface electric dipoles in the hetero stacks of SC-mTMDs, *h*-BN, and graphene layers. However, interface dipoles shift the whole energy level rigidly with respect to the vacuum level (the reference point), which does not affect individual defect states and their locations either from valence-band or conduction-band edges.

6) The dry transfer method that was used to transfer graphene to the pre-located h-BN flake on SiO₂/Si substrate should be described in details.

Response:

We regret that we did not elaborate more on the device fabrication procedures in the original manuscript. We have added extra information in the Method section, along with the device fabrication protocols for the second-type planar tunnel junction.

Added in the main text (Methods Section):

“Device Fabrication. In our planar vdW heterostructures, preparation of atomically clean interfaces is of critical importance for accurate and reliable material characterization of SC-mTMD films. At first, 60 nm to 100 nm thick high-crystalline *h*-BN flakes are mechanically exfoliated on a 90 nm thick SiO₂ layer on Si substrate. **Prior to exfoliation, we thoroughly clean the SiO₂/Si substrates with acetone and IPA in an ultrasonication bath and subsequent dipping in piranha solution. We carefully examine *h*-BN surface cleanness with a dark-filtered optical microscope to avoid cracks and non-uniform *h*-BN layers.** Then, single-layer graphene mechanically isolated onto polymer stacks of PMMA (poly(methyl methacrylate)) - PSS (polystyrene sulfonate) layers is transferred to the prelocated *h*-BN flake on SiO₂/Si substrate using a dry transfer method. **The total thicknesses of PMMA and PSS films on bare Si substrates are adjusted for optimal optical contrasts to identify the layer number of graphene and tunnel *h*-BN. Once the PMMA/PSS/Si substrates with 2D layers of graphene, *h*-BN, SC-mTMDs, and graphite on the polymer stacks are placed on top of DI water, the water soluble PSS layer is quickly dissolved and the PMMA layer with 2D materials becomes isolated from the Si substrate. Then, with a micromanipulation stage equipped with a rotator and an optical microscope, we transfer the 2D layers on PMMA to a targeted location on a SiO₂/Si substrate with micrometer accuracy. After dissolving PMMA in warm acetone (60 °C), we further anneal the samples in forming gases of Ar and H₂ (9:1 ratio by flow rate). We set the annealing temperature at 350 °C for single-layer graphene and then reduce the temperature to 250 °C for SC-mTMD films to avoid undesired defect-state formations. After annealing, we confirm the surface cleanness with an atomic force microscope.** By following a similar dry transfer protocol, SC-mTMD films, thin *h*-BN (3 to 4 layer), and graphite flakes are sequentially transferred on top of the graphite-*h*-BN stack to complete the mTMD-based planar tunneling device. **For the second type of planar tunnel devices, we intentionally align the crystalline angles of the top graphite and monolayer WSe₂. No careful alignments are necessary for transferring thin tunnel *h*-BN and underlying**

single-layer graphene films. Finally, titanium and gold (5 nm / 95 nm) electrodes are fabricated to electrically connect the top graphite and bottom graphene layers by using standard electron-beam lithography and lift-off procedures. All four high-purity (> 99.995%) SC-TMD crystals were purchased from HQ Graphene with no additional dopants added during growth procedures.”

Reviewer #3

Comments:

Jeong and coworkers report a combination of tunneling and optical spectroscopy measurements with DFT calculations to identify the band structures and defect states of four monolayer semiconducting TMDs (MoS₂, MoSe₂, WS₂ and WSe₂). By comparing bulk tunneling spectroscopy from a graphite probe, the authors conclude that S and Se vacancies are prevalent defect sources in all four materials, and are responsible for in-gap defect states.

While I believe this study has elements of current interest, I have reservations about a number of the interpretations drawn from the data presented. For this reason, I believe that publication in any form would be premature at this time. Below, I list my most pressing concerns:

Response:

We thank the Reviewer for finding our work interesting, and for the genuine concerns on our data interpretations. We regret that we originally failed to make the manuscript clear to readers, especially in the quantitative analyses of quasiparticle energy-gap sizes. We hope our detailed point-by-point responses to each question fully relieve the concerns that the Reviewer has raised.

1) There are a number of ways in which I believe that features of the tunnel spectroscopy are potentially over-interpreted (Figs. 2 and 3). The dI/dV are averaged in some way to reveal “peaks” at low bias which are said to arise from defect states. The averaging procedure is a bit mysterious to me – for example, in Fig. 2a the noise looks fairly homogeneous from 0 to -0.5 V, yet a peak somehow emerges upon averaging. How can small remaining bumps in the data be identified unambiguously as arising from in-gap defect states? Even more concerning, the graphite/graphene tunneling response itself shows unexpected peaks in dI/dV in a number of samples (Fig. S1, in particular devices 3 and 4). How can the authors distinguish between low-bias peaks they observe arising to TMD defect states and those from non-ideal tunneling into the graphene substrate?

Response:

We greatly appreciate the Reviewer’s legitimate concerns on our data interpretations, which would impact on the conclusions that we have made in the manuscript. We admit that we did not pay too much attention to the overall averaging procedures and their potential adverse implications. However, we are convinced that the averaging procedures that we have implemented barely affect the main findings of our report since all the dI/dV_b signals that we relate to chalcogen-atom vacancies and the tunnel features that we assign the key electronic structures of SC-mTMD films (such as the valence- and conduction-band edges) are more or less

insensitive to the averaging protocols. Among the defect-related dI/dV_b humps inside the energy gaps, the peak at $V_b \approx -0.26$ V as indicated with a red arrow in Fig. 2a for mWS₂ is actually the weakest signal, but the presence of such peak is not jeopardized by the mild data processing. We present a series of dI/dV_b plots for mWS₂ in the below Figure R7 while sequentially increasing the data points for averaging. As can be seen, the tunnel features at $V_b \approx -0.26$ V are distinctly identified regardless of the data points for averaging.

Figure R7 (a–f) Zoomed-in dI/dV_b plots of the mWS₂ film around the doublet e defect state while sequentially increasing the data points in the data averaging protocol.

Moreover, as we described in the main text, the dI/dV_b features relating to the SC-mTMDs inside the energy gap become immune to external gate voltage (V_g) variations thanks to a much larger charge compressibility of the bottom graphene layers. In comparison, as shown in our previous reports, tunnel signals originating from graphite/*h*-BN/graphene heterostructures become dependent on V_g since the Fermi level of graphene films is the function of external V_g . Figure R8 (b) shows a collection of dI/dV_b spectra measured from the mWS₂ device at $T = 4$ K while varying external V_g , where it is clear that the dI/dV_b hump at ≈ -260 meV does not show any V_g dependence in the range of V_g from -20 V to 20 V, while some other tunnel features change their positions when V_g varies.

Figure R8 (a) Zoomed-in dI/dV_b plot for the mWS_2 film at $V_g = 0$ V (30 points averaged). (b) A collection of dI/dV_b plots at varying V_g . (inset) Zoomed-in dI/dV_b at ≈ -260 meV.

2) I'm not sure how the gap is being extracted in these samples. For all 4 materials, the dI/dV starts to diverge from the graphene background near the valence band edge at a much lower bias than the corresponding assignment of the band edge (the deviation regions are shaded in purple). Why is this not assigned to be the valence band edge, beyond a theoretical expectation that there should be defect states residing just above the valence band? In other words, how can the authors distinguish between those two cases experimentally?

Response:

As described in the manuscript, we have not only relied on electron tunneling but also optical spectroscopy measurements to accurately assess the electronic structures of the SC-mTMD films. Besides, we have implemented DFT calculations to investigate the mid-gap states from chalcogen-atom vacancies (V_S and V_{Se}) and compared them with experiment data. In the current revision, moreover, we now additionally consider both spin-orbit coupling (SOC) and double

chalcogen-atom vacancies ($V_S V_S$ and $V_{Se} V_{Se}$) to be more accordant to our observations; please see the revised Fig. 4a and the newly added figures in the Supplementary Information. Although we attempted to clarify how to quantify the quasiparticle energy gaps and to identify key electronic structures, we admit that there existed some ambiguities in our initial description, which we hope to be resolved fully with the detailed explanations below.

At first, we quantify the valence-band splittings (Δ_{SO}) due to SOC by locating the *A* and *B* excitonic peaks through the optical reflectance and transmittance measurements (Fig. 1e and Supplementary Figures 5 and 6). Next, we assess the precise locations of each split valence-band edge in electron tunneling spectra by comparing dI/dV_b tunnel features with the optically quantified SOC-induced valence band splitting, Δ_{SO} . For example, as displayed in Fig. 2a for mWS_2 , the dI/dV_b peaks at -2.33 eV and -2.21 eV are respectively assigned at the lower SOC-split valence-band edge at K and the valence-band edge at Γ of the mWS_2 film: the valence-band edge at Γ is higher in energy than the lower SOC-split valence-band edge at K. From optical measurements with the same device, we obtained a valence-band splitting of $\Delta_{SO} \approx 0.38$ eV, which then automatically allowed us to assign the higher SOC-split valence-band edge at -1.95 eV. By following the same procedure—namely, assigning dI/dV_b spectra corresponding to the lower SOC-split valence-band edges in electron-tunneling spectroscopy measurements, and then locating the higher SOC-split valence-band edges with optically measured Δ_{SO} —we are able to extract the quasiparticle energy gaps, exciton binding energies, and other key electronic structures in all four SC-mTMD films.

Here, we would like to stress that all of our electron and optical spectroscopy measurements and DFT calculations in the current revised manuscript collectively and cohesively support our assessments on the electronic structures of SC-mTMD films, which would not be possible if we would have solely relied on either electron-tunneling or optical spectroscopy measurements alone. For example, the spectral positions where dI/dV_b start deviating from the graphene baseline (≈ 1.6 eV for mWS_2 [Fig. 2a] and ≈ 1.0 eV for $mMoSe_2$ [Fig. 3c]) would have been assigned as the top of the valence band, resulting in much lower quasiparticle energy gaps, but which would eventually fail to accommodate the Δ_{SO} from optical measurements as well as the defect states from theoretical calculations.

3) Similarly, near the conduction band the authors use deviations from the expected graphite tunnel spectra to motivate broad regions of defect states in MoS_2 and $MoSe_2$. Taking for example the MoS_2 results in Fig. 2c, there appears to be reasonably good agreement with the graphite curve shown in the green dot-dashed line even at low bias (~ -0.25 V). How can the authors be sure that the valence band edge is not at this bias instead of at -0.86 V?

Response:

We thank the Reviewer for their thoughtful judgements and genuine concerns on our data interpretation. As we attempted to diligently cover in our previous response, our assessments on the electronic structures of all four SC-mTMD planar tunnel devices were collectively made

through electron tunneling and optical (transmittance and reflectance) spectroscopy measurements, plus DFT calculations. Although some tunnel spectra, like mMoS₂ as the Reviewer pointed out, may look similar to the criteria that we set for data analysis, the resultants would simply fail to explain the other important findings. In this example, a much smaller quasiparticle energy gap of mMoS₂ would follow the assignment of the conduction-band edge at ≈ 0.25 eV, which would lead to a far smaller exciton binding energy ($E_B \leq 170$ meV) of the monolayer MoS₂ film, almost four times smaller than the binding energies of the other SC-mTMD films in the identical device scheme.

4) Assigning the band edges to the lower biases where the dI/dV first diverges from the graphite background would instead yield smaller gaps. In such a case, the interpretation of in-gap defect states would then be opposite, with little evidence for their existence. The measured dI/dV signals in the gap are very small and are being compared to an already uncertain graphite/graphene background with arbitrary multiplicative factor, so it is difficult for me to believe interpretations based on small bumps or deviations from that "background," especially those that appear only after the averaging procedure. Previous reports of the band gaps from the literature are typically smaller than the values reported here (see as just one example A. Raja et al., Nature Comm. 8, 15251 (2017)).

Response:

We appreciate the Reviewer's genuine concerns on our data interpretation. Again, our assessments made in the current manuscript on the electronic and optical structures of four SC-mTMD films are based on electron tunneling and optical spectroscopy measurements as well as DFT calculations. As discussed in the manuscript, we have paid full attention to make sure that all the experimental features and the following data assessments should be consistent in all four SC-mTMD films. Moreover, the mid-gap defect states from single chalcogen vacancies and their spectral locations inside the energy gap, singlet a_1 states close to the valence-band edge and doublet e states inside the energy gap, have been extensively investigated and well established by many others: most of them are theoretical expectations though. In our study, supported by careful DFT calculations with the considerations of SOC and double chalcogen-atom vacancies, we experimentally show that those atomic defect states possess both distinctive and common physical characteristics among the four SC-mTMD films.

It has been well known that the enhanced Coulombic interactions in atomically thin 2D layers greatly renormalize the electronic structures of SC-mTMDs, enlarging the exciton binding energies in SC-mTMDs up to ≈ 1 eV. Thus, GW calculations should be implemented to correctly address these enhanced many-electron effects, with the GW calculations known to predict substantially larger quasiparticle gaps than those from other DFT calculations without considering many-body effects. As stated in the manuscript, the large exciton binding energies of ≈ 800 meV are consistent with the enhanced Coulombic interactions, and the enhanced quasiparticle energy gaps are in perfect agreement with GW calculations for all four SC-mTMD films.

Here, we note that quasiparticle energy gaps (E_g) and exciton binding energies (E_B) are directly related with optical energy gaps (E_{opt}) by $E_B = E_g - E_{opt}$, and the optical energy gaps, the position of the A-excitonic peak, can be accurately measured in photoluminescence measurements with relatively ease. In the report by A. Raja *et al.* they also claimed that the quasiparticle energy gap and the exciton binding energy are sensitive to Coulombic interactions and nearby dielectric environments, reporting several 100s of meV changes of exciton binding energies, thereby quasiparticle energy gaps. The relatively smaller exciton binding energies can be attributed to the hydrogen-like Wannier exciton model that A. Raja *et al.* employed to extract quasiparticle energy gaps. As discussed in the report by A. Chernikov *et al.* [A. Chernikov *et al.* *Phys. Rev. Lett.* **113**, 076802 (2014)], the Wannier exciton model is known to underestimate quasiparticle energy-gap sizes in SC-TMD films.

5) The authors argue that two different WSe₂ samples exhibit qualitatively different tunnel spectroscopy owing to different rotational alignment of the graphite and WSe₂. However, the Brillouin zones of graphite and WSe₂ are very different in size owing to the large lattice mismatch, hence it is not obvious to me that rotational alignment should actually matter that much in terms of the K-point momentum offsets and subsequent tunneling constant. The authors should present a more quantitative analysis if they want to make such a claim. Furthermore, why is this only an issue in WSe₂, and not in the results of the other three TMDs?

Response:

We appreciate the Reviewer's crucial remark on our device schemes. We are aware that the Brillouin zones of the graphite probe and the WSe₂ layer cannot be aligned perfectly due to the different lattice sizes; lattice constants of graphene and WSe₂ are 0.246 nm and 0.333 nm, respectively. As we discussed in the manuscript, however, the experimentally observed spectral features from our planar tunnel junctions are signals averaged over large active areas whose dimensions are as big as several microns in width and length. Thus, there exists an area where parts of the Brillouin zones of the top graphite and the underlying SC-mTMD films are matched, and the fraction of such areas becomes maximized when the crystalline angles of the top and bottom 2D layers are perfectly aligned. Figure R9 displays optical images of all the SC-mTMD-based planar tunnel junctions discussed in the manuscript, with estimated twist angles of the SC-mTMDs and underlying single-layer graphene with respect to the top graphite probes given in each panel and summarized in the lower-right table. As indicated in Fig. R9 (d), the crystalline directions of the top graphite probe and the mWSe₂ film for the second-type device are closely aligned at $\theta_{WSe_2} \approx 0^\circ$.

Figure R9 Optical images and the twist angles of the SC-mTMD films and underlying graphene with respect to the top graphite for (a) mWS₂, (b) mMoS₂, (c) mMoSe₂, and mWSe₂ devices without (d) and with (e) angles aligned. Scale bars 20 μm.

Anyhow, we regret that we caused confusion regarding the crystalline structures of the second-type of planar tunnel devices. To explicitly mention this crystalline-angle arrangement in the momentum space, we have modified the manuscript as below. In addition, we have added Figure R9 with table in the Supporting Information.

Added in the main text. (Pages 14–15)

“... Graphite is naturally considered as the most effective tunneling probe for investigating the electronic structures of graphene and SC-mTMD films since most of the interesting features of 2D hexagonal lattice materials are confined to the electronic structures around the K point, at which the Fermi level of the graphite probe is precisely located.⁴⁹ Here, we need to assert that only parts of the Brillouin zones of the graphite probe and SC-mTMD films become matched, even for perfectly aligned planar tunnel junctions, because of the large lattice mismatch between graphene and SC-TMDs. However, the area fraction where momentum mismatch is reduced becomes maximized when the crystalline angles of the top and bottom 2D layers are perfectly aligned (Supplementary Figure 8).”

6) What is the experimental energy resolution for determining SOC splitting in the CB? All four TMDs are expected to exhibit some splitting, but a value is only reported here for WSe₂?

Response:

As stated in the manuscript, the energy resolution in our measurements is less than 10 meV, which would make it possible to experimentally observe the SOC-split conduction bands for mWS₂ and mMoSe₂, but not for mMoS₂ films. In our updated DFT calculations, we find that the conduction-band splittings due to SOCs are $\Delta_{\text{SO,C}}$ (mMoSe₂) \approx 21.3 meV, $\Delta_{\text{SO,C}}$ (mWS₂) \approx 29.7 meV, and $\Delta_{\text{SO,C}}$ (mWSe₂) \approx 33 meV, which is consistent with the experimentally quantified $\Delta_{\text{SO,C}}$ (mWSe₂) \approx 30 meV. The conduction-band splitting for mMoS₂ is found to be much smaller, $\Delta_{\text{SO,C}}$ (mMoS₂) \approx 3.2 meV. The below Table collects the SOC-induced conduction-band splittings for the four SC-mTMD films reported in several references.

Ref.	Calculation Methods.	mWSe ₂	mWS ₂	mMoSe ₂	mMoS ₂
[1]	DFT + SOC	38	27	-21	-3
[1]	TB + SOC	-3	17	-28	-4
[1]	PT	7	13	-11	-1
[2]	PT+SOC	38	26	7	3
[2]	FP+SOC	36	29	-21	-3
[3]	DFT	37	32	-21	-3
[4]	HSE,LDA	37	32	-22	-3
[4]	PBE	37	31	-20	-3
[5]	G ₀ W ₀ +BSE, HSE	16	11	-16	0
[5]	G ₀ W ₀ +PBE	-	-	11	-5
[6]	Exp.	40		-30	
	Our Observation	30	-	-	-
	Our DFT	33	29.7	21.3	3.2

(PT; perturbation theory, DFT; density functional theory, SOC; spin orbit coupling, FP; first-principles)

References:

- [1] K. Kośmider, J.W. González, J. Fernández-Rossier, Large spin splitting in the conduction band of transition metal dichalcogenide monolayers, *Phy. Rev. B* **88**, 245436 (2013).
- [2] G.-B. Liu, W.-Y. Shan, Y. Yao, W. Yao, D. Xiao, Three-band tight-binding model for monolayers of group-VIB transition metal dichalcogenides, *Phy. Rev. B* **89**, 039901 (2014).
- [3] H. Dery, Y. Song, Polarization analysis of excitons in monolayer and bilayer transition-metal dichalcogenides, *Phy. Rev. B* **92**, 125431 (2015).
- [4] A. Kormányos, G. Burkard, M. Gmitra, J. Fabian, V. Zólyomi, N.D. Drummond, V. Fal'ko, k·p theory for two-dimensional transition metal dichalcogenide semiconductors, *2D Materials*, **2**, 022001 (2015).
- [5] J.P. Echeverry, B. Urbaszek, T. Amand, X. Marie, I.C. Gerber, Splitting between bright and dark excitons in transition metal dichalcogenide monolayers, *Phy. Rev. B* **93** 121107(R) (2016).
- [6] Z. Wang, L. Zhao, K.F. Mak, J. Shan, Probing the Spin-Polarized Electronic Band Structure in Monolayer Transition Metal Dichalcogenides by Optical Spectroscopy, *Nano Lett*, **17** 740-746 (2017).

7) The figures are not referenced sequentially in the text.

Response:

We thank the Reviewer for their careful reading of our manuscript. In the revised manuscript, we have relabeled Figure 4 to be more accordant with the text.

Reviewers' comments:

Reviewer #1 (Remarks to the Author):

The updated manuscript satisfactorily addresses my comments and questions, as well as those raised by the other reviewers. I can only recommend publication of this work.

Reviewer #2 (Remarks to the Author):

I am fine with the revised manuscript and have no further comment.

Reviewer #3 (Remarks to the Author):

Jeong and coworkers have provided significant updates to their manuscript in response to my comments, as well as the comments of the other two reviewers, and as a consequence I believe their manuscript is significantly improved.

While I understand better now their reasoning used to assign band edges, I believe there are still some cases in which there is uncertainty in unambiguously interpreting tunneling spectroscopy features (in particular, the optical measurements provide a lower bound for the electronic gap, however the true conduction band minimum is difficult to uniquely define from tunnel spectroscopy features, especially in the presence of nearby defect states of initially unknown origin).

Nevertheless, I accept that this is a challenging problem, and I believe the authors have largely performed an honest accounting of the features they observe given the constraints from optical measurements and DFT (although I'm a bit skeptical how precisely electronic gaps can be calculated even at the GW level).

That said, unfortunately the authors did not address all of my comments adequately, or in some cases at all. I still have significant reservations which I would like to see addressed before I can judge suitability for publication:

1) Following my previous comment #1, I now understand and am happy with the averaging procedure. However, the authors did not address my question about the non-ideal tunneling response in their 'control' devices of graphite/graphene. I'm reproducing the question here: "the graphite/graphene tunneling response itself shows unexpected peaks in dI/dV in a number of samples (Fig. S1, in particular devices 3 and 4). How can the authors distinguish between low-bias peaks they observe arising to TMD defect states and those from non-ideal tunneling into the graphene substrate?"

2) Regarding spin-orbit splitting of the K-point valleys in the conduction band, the table the authors put together suggests that this splitting in WS₂ should be very similar to WSe₂, and that in MoSe₂ should also be roughly twice as large as the experimental energy resolution. To be more explicit, why is the spin-orbit splitting observed here only for WSe₂, but not for WS₂ and MoSe₂, which are expected to be similar? Does this suggest that the calculations are incorrect, or that the experiment is somehow not sensitive to this splitting for only certain TMDs? In any case, the authors should comment upon this.

3) I remain deeply concerned about the authors' arguments regarding the consequences of crystalline alignment between the graphite probe and the TMD.

a. Why is the tunnel spectrum for the "aligned" WSe₂ device not shown below ± 1 V bias? It would be interesting also to compare the defect states at low bias. Are the same defect states observed in both devices?

b. I appreciate the authors' inclusion of optical images to illustrate the angles between the

graphite probe and the TMD for each device, but I wonder how they know whether the identified straight edges are armchair or zigzag (or even slightly chiral)? This could introduce an uncertainty of 30° .

c. I still maintain my original concern that the large mismatch in the Brillouin zone sizes precludes the simple understanding of the tunneling response that the authors argue. The authors claim that perfect rotational alignment is required to observe enhanced tunneling between the graphite K-point and the WSe₂ K-point in Fig. 3a. However, their first WSe₂ device evidently only has a 3° rotation, and a quantitative estimate of the change in tunneling decay constant reveals that the difference between this and perfect alignment should be negligible (and in fact even large rotations of the TMD would not yield profound differences in the decay constant). I'm including a rough illustration of the Brillouin zones and Fermi surfaces for reference (not precisely to scale). The K point of graphite has momentum $\sim 1.7 \text{ \AA}^{-1}$, while the K point of the TMDs has momentum $\sim 1.3 \text{ \AA}^{-1}$. Taking a band mass of $\sim 0.5m_e$ for the TMDs, the Fermi surface has radius of $\sim 0.15 \text{ \AA}^{-1}$ a few hundred meV into the valence band, while the graphite effective mass is about an order of magnitude smaller. I'll take the extrema of the Fermi surfaces then to be separated by $\sim 0.2 \text{ \AA}^{-1}$ at perfect rotational alignment. Taking the tunneling barrier to be 3 eV for concreteness (the exact value doesn't matter), this gives a tunneling decay constant of $\sim 0.9096 \text{ \AA}^{-1}$. For a 3° rotation, a simple trigonometric estimate gives a separation of the Fermi surfaces of $\sim 0.2003 \text{ \AA}^{-1}$, and a tunneling decay constant of $\sim 0.9097 \text{ \AA}^{-1}$. Even for maximal misalignment of 30° , the tunneling decay constant is $\sim 0.917 \text{ \AA}^{-1}$. Perhaps the authors can point out a mistake in my reasoning, but the differences in decay constants as a function of twist appear to be completely negligible based on the large mismatch in the size of the TMD and graphite Brillouin zones, especially for small differences in twist angle. I remain confused then as to why the two WSe₂ devices exhibit such different responses, and wonder how consistent the results would be across different devices in general for all four TMDs investigated? Would the authors expect that two different misaligned devices yield (nearly) identical tunneling spectroscopy? If so, it would be nice to see experimental proof of this.

Reviewer(s)' Remarks to the Author:

Reviewer #3

Comments:

Jeong and coworkers have provided significant updates to their manuscript in response to my comments, as well as the comments of the other two reviewers, and as a consequence I believe their manuscript is significantly improved.

While I understand better now their reasoning used to assign band edges, I believe there are still some cases in which there is uncertainty in unambiguously interpreting tunneling spectroscopy features (in particular, the optical measurements provide a lower bound for the electronic gap, however the true conduction band minimum is difficult to uniquely define from tunnel spectroscopy features, especially in the presence of nearby defect states of initially unknown origin). Nevertheless, I accept that this is a challenging problem, and I believe the authors have largely performed an honest accounting of the features they observe given the constraints from optical measurements and DFT (although I'm a bit skeptical how precisely electronic gaps can be calculated even at the GW level).

That said, unfortunately the authors did not address all of my comments adequately, or in some cases at all. I still have significant reservations which I would like to see addressed before I can judge suitability for publication:

Response:

We thank the Reviewer for their positive comments on the revised manuscript, and agree with the Reviewer for the uphill challenges in addressing, with a high accuracy, the electronic structures of two-dimensional semiconducting films experimentally and theoretically. In that regard, we admit that our current work is yet away from the ideal metrology protocol for characterizing 2D vdW materials, but we firmly believe that our current works can provide not just a solid foundation for electronic and optical applications with layered semiconducting materials, but also many intriguing experimental schemes and theoretical problems, especially for atomic defect states in ultrathin Mo- and W-based dichalcogenide films. With that, we appreciate their genuine concerns and comments, which have made our manuscript better. Detailed point-by-point responses to each question with additional data and theoretical analyses are listed below, and all alterations to the manuscript are highlighted here in red. We hope our responses and revisions fully resolve the concerns that the Reviewer has raised.

1) Following my previous comment #1, I now understand and am happy with the averaging procedure. However, the authors did not address my question about the non-ideal tunneling response in their 'control' devices of graphite/graphene. I'm reproducing the question here: "the graphite/graphene tunneling response itself shows unexpected peaks in dI/dV in a number

of samples (Fig. S1, in particular devices 3 and 4). How can the authors distinguish between low-bias peaks they observe arising to TMD defect states and those from non-ideal tunneling into the graphene substrate?

Response:

Actually, we have addressed the above concern in our previous correspondence while answering the Reviewer’s first inquiry, as restated below.

“Moreover, as we described in the main text, the dI/dV_b features relating to the SC-mTMDs inside the energy gap become immune to external gate voltage (V_g) variations thanks to a much larger charge compressibility of the bottom graphene layers. In comparison, as shown in our previous reports, tunnel signals originating from graphite/ h -BN/graphene heterostructures become dependent on V_g since the Fermi level of graphene films is the function of external V_g . Figure R8 (b) shows a collection of dI/dV_b spectra measured from the mWS₂ device at $T = 4$ K while varying external V_g , where it is clear that the dI/dV_b hump at ≈ -260 meV does not show any V_g dependence in the range of V_g from -20 V to 20 V, while some other tunnel features change their positions when V_g varies.”

Figure R8 (a) Zoomed-in dI/dV_b plot for the mWS₂ film at $V_g = 0$ V (30 points averaged). (b) A collection of dI/dV_b plots at varying V_g . (inset) Zoomed-in dI/dV_b at ≈ -260 meV.

However, we admit that our previous response is not either clear or strong enough to draw the Reviewer's attention and eventually to resolve their concerns on our data interpretation.

As we stated in our previous response, the tunneling spectra relating to graphite-*h*-BN-graphene tunnel junctions reveal an external gate voltage (V_g) dependence because of the larger charge compressibility of graphene layers. Varying V_g shifts the Fermi level of graphene by adding additional electrons or holes, and thereby readjusting the electrostatic energy-band alignments of vertical tunnel junctions, as we discussed in detail in our previous reports [Reference #34, Jung *et al.* *Sci. Rep.* **5**, 16642 (2015), Reference #35, Jung *et al.* *Nano Lett.* **17**, 206-213 (2017), and Reference #36, Kim *et al.* *Nano Lett.* **18**, 7732-7741 (2018)]. Figure R1 (a) shows the two-dimensional representation of the tunnel spectra (dI/dV_b) in the window of V_g and V_b from one of our graphite-*h*-BN-graphene planar tunnel junctions, and Figure R1 (b) displays the collection of individual dI/dV_b plots at different V_g . It is clear that both the expected dI/dV_b relating to the Dirac point of graphene (marked with a dotted green line) and unaccountable tunnel features (marked with green arrows) changes their positions in V_b (and energy) while V_g sweeps between $V_g = -20$ V and $V_g = 20$ V.

Figures R1 (c) and R1 (d) respectively show the dI/dV_b gate mapping and the collection of individual dI/dV_b from the monolayer WS_2 planar tunnel junction (Fig. 2a). While changing V_g , there exist a few distinct dI/dV_b that change their spectra positions in V_b . Some of those features are marked with green triangles and dotted lines and we attribute them to the underlying graphene layer and its energy-band alignment with the graphite probe: detailed analyses for the origins of these tunnel features require further in-depth experimental and theoretical studies. In comparison, the dI/dV_b peak that we assign as the spectra signature for the doublet e defect states due to individual sulfur vacancies in the mWS_2 device is insensitive to V_g variation, and its spectra locations in V_b are fixed at $V_b \approx -0.26$ V, as marked with an orange triangle in Fig. R1 (d), confirming that the dI/dV_b bump at $V_b \approx -0.26$ V is not related to 'non-ideal' tunneling events into the graphene layers but the electronic structures of mWS_2 . Similarly, the dI/dV_b bumps that we assign for the mid-gap e defect states in monolayer WSe_2 films (Fig. 3a) do not change their spectra locations in V_b either, as clearly identified in the gate mapping in Fig. R1 (e) and dI/dV_b spectra plots in Fig. R1 (f).

To explicitly mention how we isolate defect-related dI/dV_b spectra, we have modified the manuscript as below. In addition, we have added Figure R1 in the Supporting Information as Supplementary Figure 8.

Added in the main text. (Pages 13)

“... In addition, an evident dI/dV_b hump inside the energy gap of mWS_2 , as marked with a red arrow and delineated in orange in Figure 2a, is assigned to the e defect states, and their spectral locations are consistent with theoretical calculations (Figure 4c). **We further note that the defect related dI/dV_b features are not sensitive to external V_g variations, unlike to tunnel spectra relating to underlying graphene and its energy-band alignment with a graphite probe (Supplementary Figure 8).** As compared with the well-isolated e defect states in mWS_2 , however, the mid-gap

states in mMoS₂ extend much wider inside the energy gap, making the defect-related tunnel spectra more conspicuous.”

Figure R1. Two-dimensional display of dI/dV_b curves at varying V_b and V_g for graphene (a), mWS₂ (c), mWSe₂ (e) based planar heterojunctions with a graphite as a tunnel probe and a thin *h*-BN as a tunnel insulator. The collections of individual dI/dV_b spectra at varying V_g are shown in (b) for the graphene, (d) for the mWS₂, and (f) for the mWSe₂ planar junctions.

2) Regarding spin-orbit splitting of the K-point valleys in the conduction band, the table the authors put together suggests that this splitting in WS₂ should be very similar to WSe₂, and that in MoSe₂ should also be roughly twice as large as the experimental energy resolution. To be more explicit, why is the spin-orbit splitting observed here only for WSe₂, but not for WS₂ and MoSe₂, which are expected to be similar? Does this suggest that the calculations are incorrect, or that the experiment is somehow not sensitive to this splitting for only certain TMDs? In any case, the authors should comment upon this.

Response:

We appreciate the Reviewer's comment on the spin-orbit splitting of SC-mTMD films around conduction-band edges. Although we agree with the Reviewer's remark that the SOC-split conduction-band minima could be observed in our WS₂ and MoSe₂ planar tunnel junctions, not just in the WSe₂ device, we would like to provide honest reservations on the possibility of their experimental observations. As we confirmed in our experiments and verified by other reports, mWS₂ and mMoSe₂ films are direct band-gap *n*-type semiconductors whose conduction-band minimum is located right at the K point. In comparison, mWSe₂ is an indirect band-gap *p*-type semiconductor and its conduction-band minimum is located at the Q and the SOC-split conduction-band minima at the K point are located at ≈ 0.15 V higher than the conduction-band edge. Thus, tunnel signals around the conduction-band minima at the K point for mWS₂ and mMoSe₂ are roughly orders of magnitude lower than those in mWSe₂ film, and the higher signal-to-noise ratio of dI/dV_b could obscure experimental identifications of those SOC-split conduction-band minima in mWS₂ and mMoSe₂. However, we might be able to beat the odds by increasing dI/dV_b in value and then improving signal-to-noise ratio through careful control of momentum mismatch of tunnel electrons in graphite-insulator-SC-mTMD planar tunnel junctions. As marked in Fig. 3a, the tunnel signals around the K point become amplified by more than one order of magnitude in value when graphite probes and mTMDs are cryptographically aligned.

3) I remain deeply concerned about the authors' arguments regarding the consequences of crystalline alignment between the graphite probe and the TMD.

- a. Why is the tunnel spectrum for the "aligned" WSe₂ device not shown below ± 1 V bias? It would be interesting also to compare the defect states at low bias. Are the same defect states observed in both devices?

Response:

In total, we have measured four different mWSe₂ planar tunnel junctions and two-dimensional displays of dI/dV_b plots for each sample are presented in Figure R2. Figures R2 (a) and R2 (b) respectively represent the 'misaligned' and 'aligned' mWSe₂ devices (Fig. 3a) in the

main manuscript. As we discussed above while answering the Reviewer’s first inquiry, the V_g -insensitive dI/dV_b bumps, thus labeled as tunnel features for the doublet e defect states in mWSe₂ films are clearly identified in the sample A (misaligned device) and also in the sample C (Fig. R2 (c)), as marked with orange triangles. However, tunnel spectra of the sample B (aligned device) below ± 1.0 V are blanked by so-called ‘unexpected’ dI/dV_b (marked as green triangles), which could be originated by the underlying graphene and its energy-band alignments with a graphite probe and obscure mid-gap states in the ‘aligned’ mWSe₂ device. Similarly, ‘unexpected’ dI/dV_b peaks are dominant in the sample D (Fig. R2 (d)), which makes us to fail to identify any mid-gap states of the mWSe₂ film.

Figure R2. Two-dimensional display of dI/dV_b curves at varying V_b and V_g for different mWSe₂ based planar heterojunctions at low temperatures, $T < 6.0$ K. dI/dV_b structures relating to the mid-gap states of mWSe₂ films are marked with orange triangles and tunnel features originated from underlying graphene are indicated with green triangles.

b. I appreciate the authors' inclusion of optical images to illustrate the angles between the graphite probe and the TMD for each device, but I wonder how they know whether the identified straight edges are armchair or zigzag (or even slightly chiral)? This could introduce an uncertainty of 30°.

Response:

We greatly appreciate the Reviewer's concerns on the possible misjudgment on the crystallographic angles of a graphite probe and SC-mTMDs films. We agree that twist-angle estimation based on optical images is not sufficient enough to identify the definite crystalline directions; armchair or zigzag, of the two layered materials. Therefore, it is plausible that the 'aligned' WSe₂ device could be misaligned by $\approx 30^\circ$ despite the crystalline angles of the top graphite and the mWSe₂ seem perfectly aligned optically. However, as we clearly state in the manuscript, the close crystallographic alignment between the graphite and the mWSe₂ in that device is not just based on the optical identification but also verified by the electrical transport measurement as shown in Fig. R3. The orange and cyan lines in Fig. R3 respectively represent $I - V_b$ and $dI/dV_b - V_b$ characteristic curves in the 'aligned' mWSe₂ device (Sample B) at $V_g = 0$ V and $T = 4.0$ K, and the negative dI/dV_b marked with brown circles, thus resonant tunneling events in the heterojunction assure that there exist points where the energy bands of graphite and mWSe₂ crisscross right around the K-point.

Figure R3. $I - V_b$ (orange curve) and $dI/dV_b - V_b$ (cyan curve) characteristic curves in the 'aligned' mWSe₂ device at $V_g = 0$ V and $T = 4.0$ K.

a. *I still maintain my original concern that the large mismatch in the Brillouin zone sizes precludes the simple understanding of the tunneling response that the authors argue. The authors claim that perfect rotational alignment is required to observe enhanced tunneling between the graphite K-point and the WSe2 K-point in Fig. 3a. However, their first WSe2 device evidently only has a 3° rotation, and a quantitative estimate of the change in tunneling decay constant reveals that the difference between this and perfect alignment should be negligible (and in fact even large rotations of the TMD would not yield profound differences in the decay constant). I'm including a rough illustration of the Brillouin zones and Fermi surfaces for reference (not precisely to scale). The K point of graphite has momentum $\sim 1.7 \text{ \AA}^{-1}$, while the K point of the TMDs has momentum $\sim 1.3 \text{ \AA}^{-1}$. Taking a band mass of $\sim 0.5m_e$ for the TMDs, the Fermi surface has radius of $\sim 0.15 \text{ \AA}^{-1}$ a few hundred meV into the valence band, while the graphite effective mass is about an order of magnitude smaller. I'll take the extrema of the Fermi surfaces then to be separated by $\sim 0.2 \text{ \AA}^{-1}$ at perfect rotational alignment. Taking the tunneling barrier to be 3 eV for concreteness (the exact value doesn't matter), this gives a tunneling decay constant of $\sim 0.9096 \text{ \AA}^{-1}$. For a 3° rotation, a simple trigonometric estimate gives a separation of the Fermi surfaces of $\sim 0.2003 \text{ \AA}^{-1}$, and a tunneling decay constant of $\sim 0.9097 \text{ \AA}^{-1}$. Even for maximal misalignment of 30°, the tunneling decay constant is $\sim 0.917 \text{ \AA}^{-1}$. Perhaps the authors can point out a mistake in my reasoning, but the differences in decay constants as a function of twist appear to be completely negligible based on the large mismatch in the size of the TMD and graphite Brillouin zones, especially for small differences in twist angle. I remain confused then as to why the two WSe2 devices exhibit such different responses, and wonder how consistent the results would be across different devices in general for all four TMDs investigated? Would the authors expect that two different misaligned devices yield (nearly) identical tunneling spectroscopy? If so, it would be nice to see experimental proof of this.*

Response:

We appreciate the Reviewer's sincere comments to verify the concreteness of our understanding on the momentum angle-dependent tunneling spectra. We feel indebted to their genuine inquiries for leading us to analyze the 2D vdW-based planar heterojunctions and the charge transport behaviors through the heterostructures with great care, which we find interesting and worthy for further in-depth experimental and theoretical studies. Hopefully, we will be able to report a full detailed analysis on this emerging interesting topic of vertical quantum transport in ultrathin 2D vdW heterostructures in coming literature.

As for the beginning and to relieve the Reviewer's concerns regarding this manuscript in the first place, we have conducted a numerical simulation of quantum tunneling events through a graphite probe and mTMD vertical heterojunctions with an *h*-BN insulating layer. Although the Reviewer's evaluations on the tunneling decay constants considering the first Brillouin zones (BZ) of graphite and mWSe₂ separately seem flawless, we would like to point out that there are a couple of critical points required to apply the Reviewer's argument to our 2D vdW planar tunnel junctions. (i) The first point is that we need to consider extended BZs as well as the 1st BZ of the heterostructures since newly formed 2D vdW heterojunctions consisting of 2D layered materials

with different lattice constants (such as graphite and mTMDs) have a new lattice periodicity, often cited as moiré superlattices. (ii) The second point is that the equipotential surfaces of graphite and mTMDs in energy-momentum spaces where charged carriers are elastically tunneled become V_b -dependent, and often larger, when the size of the potential surfaces matter because we need to impose higher V_b to inject electrons to either conduction or valence bands of the SC-mTMDs.

In the following, we discuss the first point with a simplified toy model, and conclude that it is essential to take into account crystalline momentum alignments within the first BZ of the moiré superlattices; this is equivalent to the extended BZs of subsystems, rather than exclusively dealing with the first BZs of individual subsystems (graphite or mTMDs). Next, we apply the developed toy model to our planar heterojunctions of graphite probe and mWSe₂ layers while taking the second point into consideration, and prove that electron tunneling is indeed maximized when individual subsystems are crystallographically aligned perfectly.

To succinctly provide the physical contents of the first point, we begin with a simple toy model consisting of two one-dimensional systems as schematically illustrated below in Figure R4.

Figure R4. Diagram of a double-layer vdW heterostructure. Tunnel electrons are injected at the bottom monitored through the top layer.

The lattice constants of systems 1 and 2 are considered as a_1 and $a_2 = 1.5 \times a_1$, respectively, and the BZs of the individual subsystems 1 and 2 are represented as boxes in Figure R5.

Figure R5. Brillouin zones of the subsystems 1 and 2 in Fig. R1.

When we consider the first BZs of subsystems 1 and 2, resonant tunneling occurs at the positive energy (marked with a red arrow in Fig. R5). However, the superlattice comprising subsystems 1 and 2 has a new lattice periodicity of $a_0 (= 3a_1 = 2a_2)$, and the first BZ of the heterostructure becomes smaller and provides an additional point at the negative energy (marked with a blue arrow in Figure R6) where a second resonant tunneling occurs. The point where the second resonant tunneling happens is equivalently located in the extended BZ scheme, as shown in Figure R7. We note that the equivalent extended BZ requires a number of higher-order BZs, with which the band structures of two subsystems become periodic.

Figure R6. Energy-momentum dispersion curve in the 1st Brillouin zone of the heterostructure with periodicity a_0 .

Figure R7. Extended Brillouin zone of the heterostructure, which is equivalent to Figure R3.

To additionally confirm the secondary resonant tunneling event in the extended BZs of 2D vdW heterostructures, we have conducted numerical simulations of tunneling events through subsystems 1 and 2 in the structure of planar heterojunctions. As shown in Figure R8, the secondary resonant tunneling occurs at the negative energy at which the crystal momentums of the two systems are aligned in the second BZ, confirming that we need to consider not only the first BZ but also extended BZs to fully account for the tunneling events in 2D vdW planar tunnel junctions. For the numerical simulations, we implement a recursive Green function method based on the lattice model and calculate the electron-tunneling transmission probabilities for the case of incoming electrons from subsystem 2 tunneling into the outgoing modes of subsystem 1. The energies where the resonant tunneling events occur are consistent with those extracted from the simple toy model.

- 1) The energy where Resonant tunneling occurs matches exactly to the above figure.
- 2) The resonant tunneling at *the second Brillouin zone* happens indeed.

Figure R8. Numerical simulation of tunneling events in the double-layer vdW heterostructure in Fig. R1.

Now we apply the developed toy model to our planar tunnel junctions of graphite probes and SC-mTMDs. The diagrams in Figure R9 show the extended BZs of graphite and mWSe₂ with K and K' points respectively marked as black and red dots. The left and right panels represent graphite-mWSe₂ heterojunctions with twist angles of 0° and 3°, respectively. As the Reviewer correctly pointed out, it is found that the tunnel decay constants are not sensitive to the twist angle when the first BZ is solely compared. However, the tunneling rate becomes highly dependent on the misalignment angle when we extend our discussions to the extended BZs. It is

Figure R9. Extended BZs of graphite and mWSe₂ with K and K' points respectively marked as black and red dots for heterostructures with (left) 0° and (right) 3° twist angles.

found that the momentum mismatch is significantly modified in the extended BZs, as we note from the K and K'-points in the 3rd and 4th BZs in Figure R9.

Moreover, the size of the extended BZs becomes highly dependent on the twist angle. Figure R10 shows the moiré pattern in momentum space of graphite and WSe₂ for twist angles of 0°, 3°, 30°, and 33°, where the size of the extended BZs marked with a solid black rhombus shrinks as the twist angle increases. We consider these twist angle-dependent extended BZ sizes as main attributes for the enhancement of tunnel dI/dV_b in perfectly aligned 2D vdW heterostructures with graphite and TMDs, as we deliberate below.

Figure R10. Gray solid circles show K and K'-points of graphite, while red and blue are for those of TMD. Yellow color shows Γ -point. Moiré pattern arises with 0°, 3°, 30°, and 33° twist angles (from left).

As mentioned in the beginning, we need to consider the varying equipotential surfaces of graphite and SC-mTMDs in energy-momentum spaces to accurately assess electron tunneling in the 2D planar tunnel junctions. Figure R11 displays energy-momentum dispersion curves of graphite (solid black line) and mWSe₂ (solid red line) around the K point. Even at low energies, we can find that a fairly large separation of the equipotential surfaces exists, as marked with a dotted blue line in Figure R11. Under non-zero bias voltages, however, tunnel electrons have larger momentum and a point where the two bands cross and resonant tunneling occurs is formed, as marked with a dotted green line. Figure R12 illustrates the extended BZs with the equipotential surfaces of graphite and WSe₂ at varying bias voltages from 1.2 V to 2 V for the graphite-WSe₂ heterostructures with twist angles 0° and 3°.

Figure R11. Energy-momentum dispersion curves of the graphite and mWSe₂.

Figure R12. Extended BZs marked with the equipotential surfaces of graphite and mWSe₂ with twist angles of (left) 0° and (right) 3°.

Fully incorporating the first (i) and the second (ii) points that the Reviewer honestly missed, we have calculated the tunneling decay constants (T) around the K points and applied them to the following equation, $\exp(-T \times \Delta z)$, directly relating to experimentally observable dI/dV_b . Here, Δz is the average distance between the nearest atoms of graphite and mWSe₂. Below, Figure R13 displays a log plot of the summation of the tunneling decay constants integrated around the K points at varying twist angles of 0°, 3°, 30°, and 33°. We use $\Delta z = 4 \times 3.25 \text{ \AA}$ for all twist angles (3.25 Å is the thickness of the *h*-BN monolayer), and consider the effective masses of graphite and mWSe₂ as those suggested by the Reviewer. As confirmed below, the simulated dI/dV_b across the graphite-*h*-BN-mWSe₂ planar tunnel junctions becomes highest when the top graphite and the mWSe₂ layers are crystallographically aligned, consistent with the results in our current manuscript.

Figure R13. Simulated dI/dV_b curves as a function of V_b at different twist angles: 0° (blue), 3° (red), 30° (yellow), and 33° (purple). The y-axis is drawn with an arbitrary unit.

To explain newly found angle-dependent tunneling events through 2D vdW vertical heterostructures, we have modified the manuscript as below.

Added in the main text. (Pages 15)

“ Here, we need to assert that only parts of the Brillouin zones (BZ) of the graphite probe and SC-mTMD films become matched, even for perfectly aligned planar tunnel junctions, because of the large lattice mismatch between graphene and SC-TMDs. However, electron tunneling through 2D vdW heterojunctions consisting of layered materials with different lattice constants needs to consider extended BZs of the heterostructures and the varying equipotential surfaces of graphite and SC-mTMDs depending on V_b . We find that both of which are sensitive to the misalignment angle of the junctions and tunneling probability becomes highest for perfectly aligned tunnel junctions. We defer detailed discussions on the charge transport through vertical 2D vdW heterostructures in coming literature.”

REVIEWERS' COMMENTS:

Reviewer #3 (Remarks to the Author):

The authors have resolved my remaining concerns and strengthened their manuscript. I believe it is now suitable for publication in Nature Communications.